EMBO
Molecular Medicine

# Blocking sense-strand activity improves potency, safety and specificity of anti-hepatitis B virus short hairpin RNA

Thomas Michler[1,2], Stefanie Große[3,4], Stefan Mockenhaupt[3,4,†], Natalie Röder[1], Ferdinand Stückler[5], Bettina Knapp[5], Chunkyu Ko[1], Mathias Heikenwalder[1,‡], Ulrike Protzer[1,2,*,§] & Dirk Grimm[3,4,**,§]

## Abstract

Hepatitis B virus (HBV) is a promising target for therapies based on RNA interference (RNAi) since it replicates via RNA transcripts that are vulnerable to RNAi silencing. Clinical translation of RNAi technology, however, requires improvements in potency, specificity and safety. To this end, we systematically compared different strategies to express anti-HBV short hairpin RNA (shRNA) in a pre-clinical immunocompetent hepatitis B mouse model. Using recombinant Adeno-associated virus (AAV) 8 vectors for delivery, we either (i) embedded the shRNA in an artificial mi(cro)RNA under a liver-specific promoter; (ii) co-expressed Argonaute-2, a rate-limiting cellular factor whose saturation with excess RNAi triggers can be toxic; or (iii) co-delivered a decoy ("TuD") directed against the shRNA sense strand to curb off-target gene regulation. Remarkably, all three strategies minimised adverse side effects as compared to a conventional shRNA vector that caused weight loss, liver damage and dysregulation of > 100 hepatic genes. Importantly, the novel AAV8 vector co-expressing anti-HBV shRNA and TuD outperformed all other strategies regarding efficiency and persistence of HBV knock-down, thus showing substantial promise for clinical translation.

**Keywords** Adeno-associated virus; hepatitis B virus; RNA interference; short hairpin RNA; tough decoy

**Subject Categories** Genetics, Gene Therapy & Genetic Disease; Microbiology, Virology & Host Pathogen Interaction

## Introduction

Chronic hepatitis B caused by hepatitis B virus (HBV) infection remains a major health burden that affects nearly 250 million humans and causes an estimated 686,000 annual deaths from the long-term consequences liver cirrhosis and hepatocellular carcinoma (HCC) (MacLachlan *et al*, 2015; Schweitzer *et al*, 2015). This is due to the limitations of standard-of-care antiviral therapy, including that nucleos(t)ide analogs require daily and lifelong application, which is costly, may elicit side effects such as kidney insufficiency (Tenofovir and Entecavir) and can prompt the emergence of resistant HBV mutants (Song *et al*, 2012). Moreover, current interventions fail to eliminate the virus or to inhibit viral transcription and translation, despite the ability of these drugs to control the generation of infectious HBV virions. However, the viral proteins crucially determine viral persistence and pathogenesis. In particular, secreted HBV surface (HBsAg) and e antigen (HBeAg) are proposed to be key players in modulating the host immune response that impair its ability to clear the virus (Protzer *et al*, 2012), and that very likely contribute to development of HBV-associated HCC (Ringelhan *et al*, 2013). HBV X protein (HBx) can affect immune signalling and contributes to HCC development (Levrero & Zucman-Rossi, 2016). Furthermore, HBV core (HBc) and polymerase initiate the first step of replication, encapsidation and reverse transcription of pregenomic HBV RNA.

A possible solution towards the development of new and more effective HBV therapies is RNA interference (RNAi). Particularly encouraging is that RNAi allows suppression of all viral transcripts and proteins and may serve as a basis to restore immune responses. This was first suggested a decade ago when it was reported that expression of shRNAs (short hairpin RNAs, artificial triggers of RNAi) can be harnessed for *in vivo* inhibition of HBV or hepatitis C

1 Institute of Virology, Technische Universität München/Helmholtz Zentrum München, München, Germany
2 German Center for Infection Research (DZIF), partner site München, München, Germany
3 Department of Infectious Diseases/Virology, Cluster of Excellence CellNetworks, Heidelberg University Hospital, Heidelberg, Germany
4 BioQuant, University of Heidelberg, Heidelberg, Germany
5 Institute of Computational Biology, Helmholtz Zentrum München, Neuherberg, Germany
   *Corresponding author. Tel: +49 89 4140 6821; E-mails: protzer@tum.de; protzer@helmholtz-muenchen.de
   **Corresponding author. Tel: +49 6221 5451339; E-mail: dirk.grimm@bioquant.uni-heidelberg.de
   †Present address: School of Biological Sciences, Nanyang Technological University, Singapore City, Singapore
   ‡Present address: Division of Chronic Inflammation and Cancer, German Cancer Research Center (DKFZ), Heidelberg, Germany
   §These authors contributed equally to this work

virus (HCV) gene expression (McCaffrey *et al*, 2002, 2003). This pioneering work was quickly followed by a series of *in vitro* and *in vivo* studies in pre-clinical HBV models that verified the enormous promise of RNAi therapeutics for HBV treatment (McCaffrey *et al*, 2003; Carmona *et al*, 2006; Grimm *et al*, 2006, 2010; Giering *et al*, 2008; Keck *et al*, 2009), and that culminated in a phase II clinical evaluation of a polyconjugated siRNA in chronically HBV-infected patients (www.arrowheadresearch.com/programs/ARC-520). This inspiring translation from bench to bedside was fuelled by several facts that make RNAi highly attractive and promising as an antiviral modality. Firstly, the four major HBV transcripts share a common 3′ end which allows concurrent targeting of all viral messenger (m)RNAs with a single RNAi molecule (Ebert *et al*, 2011). Secondly, HBV mRNAs are constantly transcribed from the viral covalently closed circular DNA (cccDNA), the persistent HBV form in the nucleus of infected hepatocytes. This renders HBV highly vulnerable to a gene therapy approach using a vector that permanently expresses an effective anti-HBV shRNA directed against the 3′ end of all HBV RNAs. One exciting option to deliver shRNAs into the infected hepatocyte are recombinant Adeno-associated viruses of serotype 8 (AAV8), non-pathogenic vectors that mediate efficient and stable hepatic gene transfer in small and large animals as well as in humans (Grimm *et al*, 2006; Chen *et al*, 2007, 2008; Giering *et al*, 2008; Nathwani *et al*, 2011). Increasing their appeal for treatment of persistent HBV infection is our recent observation that the HBV X protein enhances nuclear AAV transport and thus "helps" AAV transduction, indicating that the vector will preferentially establish in HBV-infected hepatocytes (Hosel *et al*, 2014).

To fully realise the potential of AAV/RNAi vectors for HBV therapy and to foster their clinical translation, it is now imperative to overcome lingering safety concerns (Grimm *et al*, 2006; Grimm, 2011). These were initially triggered by observations with first-generation vectors that mediated overly abundant shRNA expression, causing liver toxicity with elevated serum liver transaminases, jaundice, weight loss, as well as histological and structural liver alterations. In extreme cases, unrestricted shRNA over-expression even resulted in organ failure and morbidity of treated mice (Grimm *et al*, 2006). Additional findings in c-Myc-transgenic mice indicate that shRNA over-expression can also accelerate tumorigenesis under certain conditions (Beer *et al*, 2010). Notably, adverse effects from shRNA over-expression were likewise observed in other species and tissues beyond the mouse liver (Grimm, 2011).

Fortunately, we and others could identify possible mechanisms underlying this RNAi toxicity, which now offers various avenues for improvement (Fig 1A and B). One model suggests that ectopic RNAi triggers overwhelm the endogenous mi(cro)RNA pathway, particularly Exportin-5 and Argonaute-2 (Ago-2, key RISC component) and thus perturb miRNA biogenesis and/or activity in a dose-dependent manner (Grimm *et al*, 2006, 2010; Giering *et al*, 2008). Evidence is that RNAi efficiency can be enhanced by Ago-2 co-expression *in vitro* and *in vivo* (Diederichs *et al*, 2008; Grimm *et al*, 2010; Borner *et al*, 2013). Moreover, the use of weaker shRNA promoters (Giering *et al*, 2008; Grimm *et al*, 2010; Suhy *et al*, 2012) or shRNA embedding in a natural miRNA context can reduce accumulation of mature RNAi triggers and thus increase safety (Zeng *et al*, 2005; McBride *et al*, 2008; Boudreau *et al*, 2009; Ely *et al*, 2009), albeit even marginal shRNA expression can provoke phenotype and transcriptome changes in mouse livers (Maczuga *et al*, 2014).

A second model implies that RNAi triggers perturb cell homoeostasis through unwanted inhibition of "off-target" genes with partial complementarity to one of the two arms of the double-stranded RNAi molecule (Jackson *et al*, 2003; Scacheri *et al*, 2004; Fedorov *et al*, 2006). As vector-encoded shRNAs are incompatible with molecular or chemical modifications that may improve specificity (Grimm, 2009), the best remaining option is judicious selection of shRNAs with an inherent bias towards the antisense arm that directs RISC to the target mRNA (Khvorova *et al*, 2003; Reynolds *et al*, 2004; Gu *et al*, 2014; Liu *et al*, 2015). However, latest data show that shRNA off-targeting can also originate from binding of cellular RNA sequences by the sense strand, which is identical to the target region (Clark *et al*, 2008; Wei *et al*, 2009; Kwak & Tomari, 2012; Schurmann *et al*, 2013; Gu *et al*, 2014; Mockenhaupt *et al*, 2015). To block this side effect, we recently (Mockenhaupt *et al*, 2015) developed a novel bi-cistronic AAV vector co-expressing a shRNA with a second RNA hairpin called "tough decoy" or "TuD". Originally devised by Haraguchi and colleagues to silence cellular miRNAs (Haraguchi *et al*, 2009), we found that TuDs can be repurposed to selectively bind and inactivate shRNA sense strands, thereby improving RNAi specificity and promising a crucial benefit for clinical RNAi therapies. In this study, we systematically compared our new shRNA/TuD design side by side to two other advanced shRNA expression strategies—Ago-2 co-delivery and embedding in a miRNA scaffold—for HBV inhibition in a transgenic mouse model of chronic HBV infection.

# Results

### Design of different RNAi triggers for direct *in vitro* and *in vivo* comparison

Figure 1C provides a schematic overview of the four distinct RNAi expression strategies that were tested in this study. As miRNA scaffold for embedding of the anti-HBV shRNA (strategy [ii]), we selected miR-122 based on its high and specific expression in the liver (Landgraf *et al*, 2007). The final construct was designed to fulfil two requirements (Fig EV1A top): (i) the antisense strand is in the 5′ position, to mimic the miR-122 structure; and (ii) the first position of the predicted pre-miRNA (processed pri-miRNA) is a uridine, to recapitulate the thermodynamic stability of the first bulge of pre-miR-122. In contrast, rules for a construct expressing a conventional anti-HBV shRNA (Figs 1Ci and EV1A bottom) were (i) the sense strand is in the 5′ arm to facilitate neutralisation by a co-expressed TuD (Mockenhaupt *et al*, 2015); and (ii) this strand starts with guanine, to allow precise transcription from RNA polymerase III promoters. For shRNA expression, we used the H1 promoter based on our experience that it yields safer and more stable RNAi in mouse livers than the stronger U6 promoter (Grimm *et al*, 2010).

To select anti-HBV shRNAs that could be modified according to these rules, we focused on the X region of the HBV genome which enables simultaneous targeting of all viral transcripts (Ebert *et al*, 2011) (Appendix Fig S1A and B). We identified four shRNA candidates (Fig EV1B) and measured their efficacy in an HBV cell culture model (Fig 2). From this pre-screen, we chose shHBV7 for all further studies as it consistently gave best results, including 99%

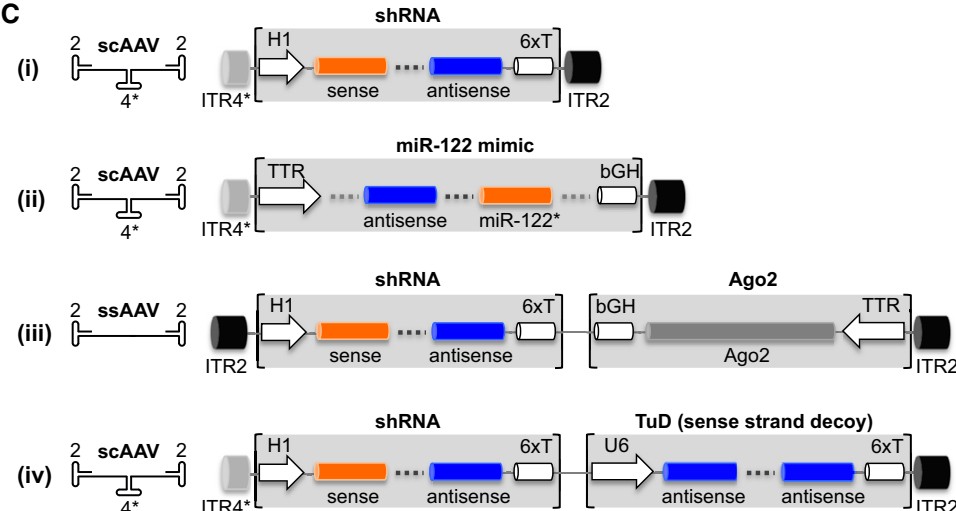

**Figure 1.  Mechanisms that can cause RNAi toxicity and strategies to overcome them.**

A   Three possible mechanisms explaining toxicity after over-expression of conventional shRNAs (see text for details). Orange, shRNA sense strand; blue, shRNA antisense strand.

B   Improved shRNA expression strategies to circumvent the mechanisms in panel (A).

C   Four principal vector designs compared in this study. Symbols on the left indicate the AAV vector backbone; numbers denote serotype origin of the ITRs (inverted terminal repeats, serving as AAV DNA packaging signals; asterisk indicates a deletion in the AAV4 ITR creating the self-complementary [sc]AAV genotype). Six thymines (6xT) serve as termination signal for H1/U6 promoters; bGH, bovine growth hormone polyadenylation signal; TTR, transthyretin promoter.

knock-down of secreted HBsAg (measured by a quantitative HBsAg chemiluminescent microparticle immunoassay [CMIA], Fig 2A), as well as robust knock-down of secreted HBeAg and HBV DNA, and of intracellular pre-genomic HBV RNA (3.5-kb transcript), total HBV RNA and HBV DNA (Fig 2A–D). Notably, the used HBeAg test (Fig 2A) does not measure true concentrations, but a signal-to-control ratio (S/CO) which follows a sigmoid curve. This leads to underestimates at high concentrations and, *vice versa*, to overestimates at low concentrations. Accordingly, the HBeAg reductions are likely even more pronounced and in the high range measured with the quantitative HBsAg test. Moreover, an additional benefit of shHBV7 is that its target sequence is largely conserved across all HBV genotypes (Appendix Fig S1C).

We next embedded this lead shRNA into the pre-miR-122 scaffold, following the scheme in Fig EV1C. Therefore, we added

two uridines to the 3′ end of the miR-122/HBV7 (from hereon called miHBV7) antisense strand to match the conventional shRNA [expected to carry a 3′ overhang of up to three uridines, typical for shRNAs expressed from RNA polymerase III promoters (Cullen, 2006)] (Fig EV1C top). Moreover, we designed a partially complementary sense strand mimicking the mismatches and bulges in genuine miR-122 and finally added 5′ and 3′ flanking sequences from miR-122 to complete the miHBV7 scaffold (Fig EV1C bottom). The entire sequence was then cloned into an AAV vector and expressed from the liver-specific RNA polymerase II transthyretin promoter (TTR) (Wu *et al*, 2008).

This promoter was also used to co-express Ago-2 together with H1 promoter-driven shHBV7 (Fig 1Ciii). The large size of the bi-cistronic shRNA/Ago-2 construct required cloning into a traditional AAV vector which packages as single-stranded DNA.

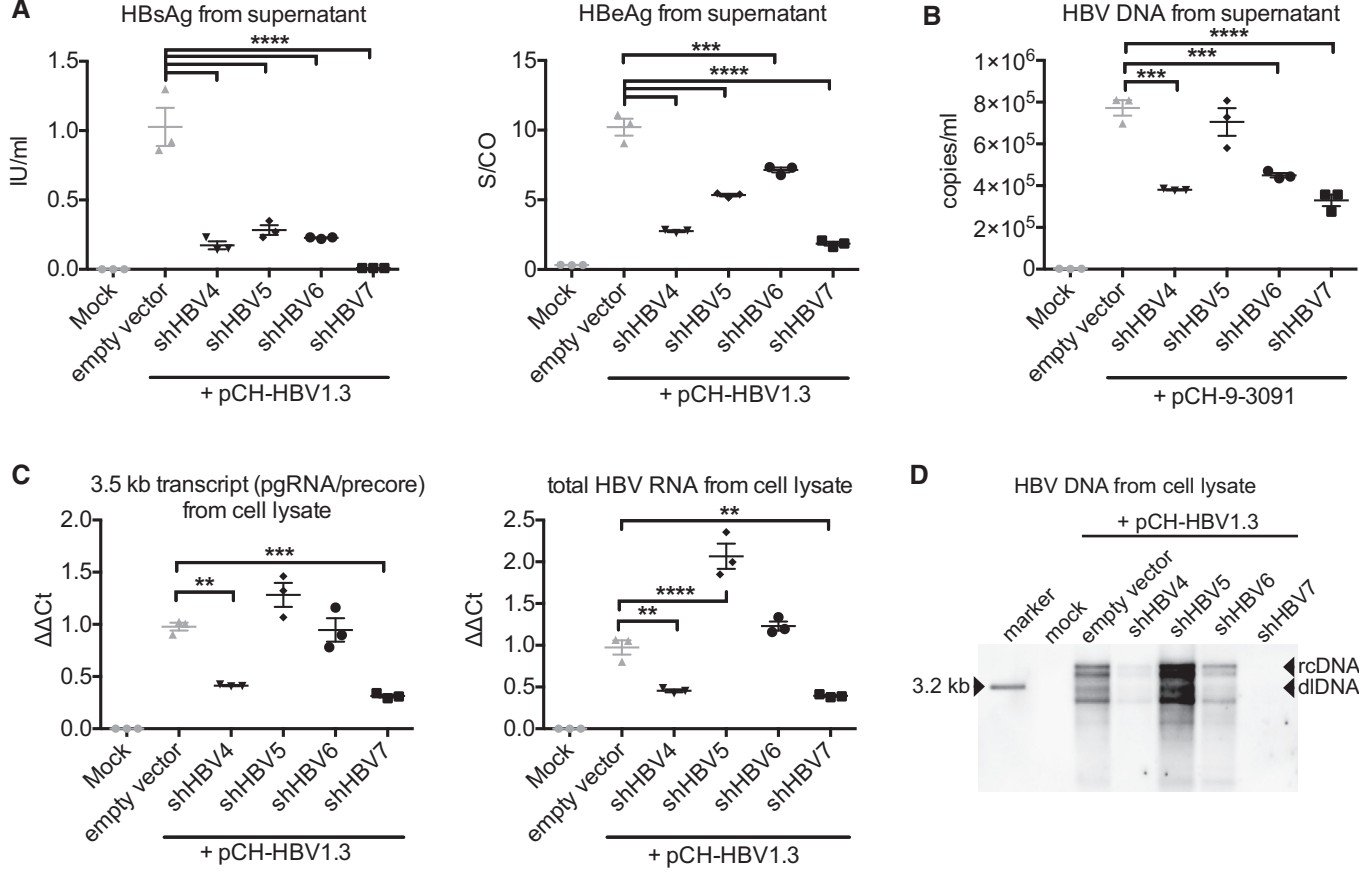

**Figure 2. Pre-selection of anti-HBV short hairpin RNAs.**

A  Huh7 cells were co-transfected with shRNAs fulfilling all requirements for incorporation into the expression strategies from Fig 1C and the HBV expression plasmid pCH-HBV1.3, and HBeAg and HBsAg were quantified in the supernatant 48 h later. An empty AAV vector plasmid was used as control. Another negative control was mock-transfected cells.

B  The same experiment was repeated with plasmid pCH-9-3091, which contains a 1.1-fold over-length HBV genome. HBV DNA in supernatant was measured at day four via PCR, using primers that preferentially amplify HBV rcDNA rather than the 1.1-fold over-length HBV sequence contained in the plasmid.

C  RT-qPCR results from cell lysates (same experiment as in panel A) with primers specific for the 3.5-kb HBV transcripts (consisting of pre-genomic [pg] RNA and the pre-core transcript; left), or with primers that detect all HBV transcripts (right).

D  Southern blot analysis of lysates obtained 72 h post-transfection of HepG2 cells treated with the same plasmids as in panel (A).

Data information: Diagrams in panels (A–C) show mean values and SEM (all experiments were performed in triplicates). Significance was calculated using one-way ANOVA with Tukey's multiple comparison correction. S/CO, signal-to-control ratio; rcDNA, relaxed circular DNA; dlDNA, duplex-linear DNA, double-stranded linear DNA; n.s., non-significant; *$P < 0.05$; **$P < 0.01$; ***$P < 0.001$; ****$P < 0.0001$. See Appendix Table S1 for exact $n$- and $P$-values.

Conversely, the other two cassettes—H1-shHBV7 or TTR-miHBV7— were assembled in a self-complementary (sc)AAV-2/4 vector genome (Grimm *et al*, 2006). These differences are irrelevant for transfection experiments in cell culture but essential *in vivo* where scAAV vectors express more rapidly and robustly (Grimm *et al*, 2006; McCarty, 2008).

**TuD inclusion to reduce shRNA off-targeting and improve RNAi efficiency**

Next, we engineered a TuD RNA hairpin to specifically bind and neutralise the shHBV7 sense strand (Fig 1Civ). Based on our recently elucidated rules (Mockenhaupt *et al*, 2015), we cloned two perfect binding sites for the shHBV7 sense strand and expressed the resulting TuDHBV7 from a U6 promoter. The U6-TuD was then

inserted downstream of the H1-shHBV7 cassette in the scAAV vector backbone (Appendix Fig S2). To assess shHBV7 sense versus anti-sense-strand activity and TuD functionality, we co-transfected cells with the shHBV7/TuDHBV7 plasmid and dual-luciferase reporters carrying a perfect binding site for either shHBV7 sense or antisense strand in the 3′ untranslated region (UTR) of *Renilla* luciferase (Fig 3A). Negative controls included (i) an irrelevant shRNA against human alpha-1-antitrypsin (shα1AT), (ii) a TuD against the sense strand of shα1AT and (iii) a luciferase reporter without shRNA binding sites.

Expression of shHBV7 substantially inhibited the antisense-strand reporter, irrespective of which TuD was co-expressed (Fig 3B, shHBV7 group). The sense-strand reporter was also suppressed by > 65% in the presence of the control TuD, solidifying our recent data that both shRNA strands are frequently active

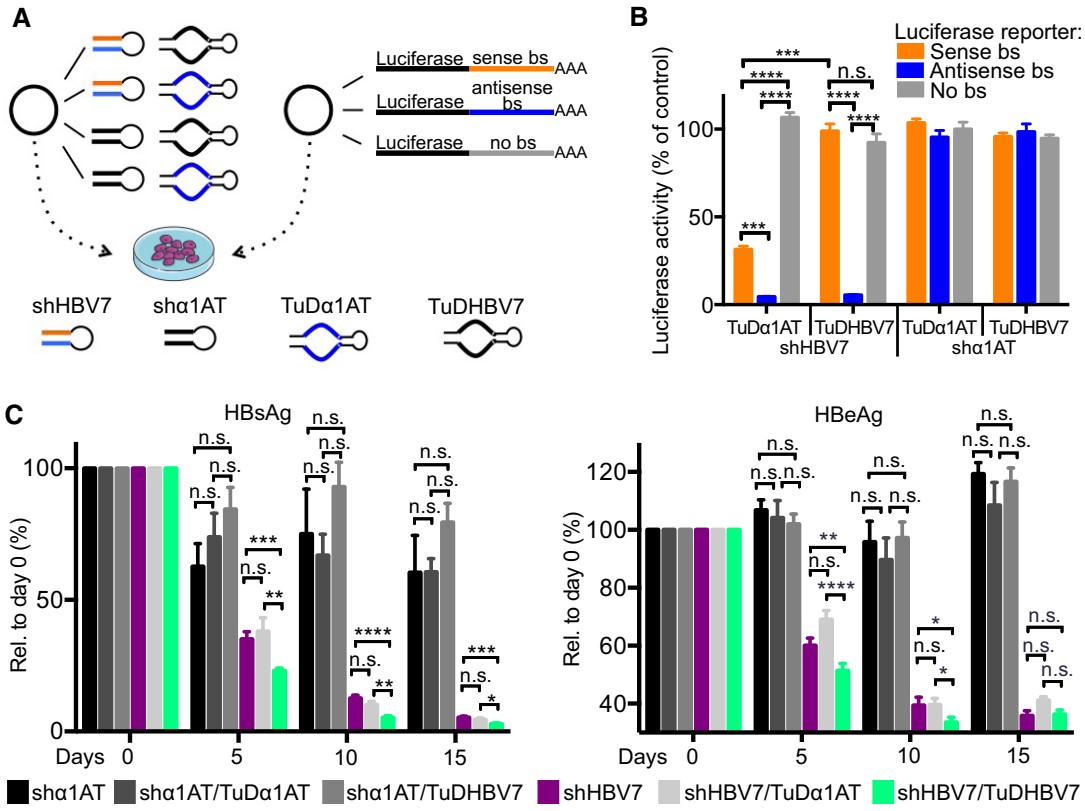

**Figure 3.  Improvement of shHBV7 specificity and efficiency by co-expression of a TuD against the sense strand.**

A  Scheme of the assay to determine shRNA strand activities in the absence or presence of specific TuDs. Huh7 cells were co-transfected with the shRNA and the TuD encoded on the same plasmid, and a luciferase reporter with appropriate shRNA strand binding sites. bs, binding site.

B  Measurement of resulting luciferase activities in cell lysates 48 h post-transfection. Note the potent inhibition of shHBV7 sense-strand activity with the specific TuDHBV7, without interference with the antisense strand (compare left two orange or blue bars, respectively). bs, binding site.

C  *In vivo* evaluation of different shRNA and TuD combinations after packaging into double-stranded AAV8 vectors and intravenous (i.v.) injection of $1 \times 10^{11}$ particles into HBV-transgenic mice (HBV1.3.32). Anti-HBV activity was determined by measuring HBsAg and HBeAg levels in the serum of treated animals at the indicated time points post-AAV injection.

Data information: Diagrams in panels (B and C) show mean values and SEM. Significance was calculated using Student *t*-test. The experiment in panel (B) was performed in triplicates. The diagram in panel (C) shows pooled data of three experiments with at least five mice per treatment group. n.s., non-significant; \*$P < 0.05$; \*\*$P < 0.01$; \*\*\*$P < 0.001$; \*\*\*\*$P < 0.0001$. See Appendix Table S1 for exact *n*- and *P*-values. Panel (A) contains clipart from Servier Medical Art.

(Mockenhaupt *et al*, 2015). Importantly, this unwanted sense-strand activity was completely abolished upon co-expression of TuDHBV7 (Fig 3B, shHBV7 group). All controls behaved as expected, since expression of the shHBV7-sensitive luciferase reporter was neither inhibited by the irrelevant shRNA nor by the two TuDs (which exert no inherent RNAi activity).

We next aimed to verify these results in an *in vivo* model of chronic hepatitis B. For this, we used an HBV-transgenic mouse strain HBV1.3.32, which carries a chromosomally integrated 1.3-fold over-length HBV genome (Guidotti *et al*, 1995). This viral DNA expresses all HBV transcripts and antigens, and it is replication-competent and produces infectious wild-type HBV genotype D. We packaged the various constructs encoding only shRNA or shRNA plus TuD (Fig 3C, indicated at the bottom) into AAV8 capsids and intravenously injected $1 \times 10^{11}$ particles per mouse. Remarkably, the dual vector co-expressing shHBV7 and TuDHBV7 not only efficiently suppressed HBV but outperformed the two vectors expressing shHBV7 alone, or co-encoding shHBV7 and the control TuD, as

evidenced by significantly greater reductions of serum HBsAg and HBeAg (Fig 3C, green bars). The difference was particularly pronounced at the earliest time point (5 days) post-injection, showing that the shRNA7/TuD construct is also faster at inhibiting HBV *in vivo*. The control shRNA only had a minor effect on HBsAg expression (black/grey bars) and neither HBsAg nor HBeAg were significantly altered upon co-expression of either TuD, arguing against a general influence of TuDs on HBV expression.

We subsequently compared all four vector variants from Fig 1C side by side in the HBV cell culture model for their effects on HBV proteins, DNA and RNA (Fig 4). Despite the high inherent activity of the shHBV7 shRNA, we observed a trend towards further improvement through Ago-2 co-expression at least for the secreted parameters (best notable for HBV DNA in the supernatant, whose knock-down was increased from 57.3 to 73.5%). This is in line with the rate-limiting nature of Ago-2 described by us and others (Diederichs *et al*, 2008; Grimm *et al*, 2010; Borner *et al*, 2013). The TTR-miHBV7 construct was typically less efficient than the other

strategies but still achieved substantial anti-HBV RNAi. Southern blot analysis revealed a robust reduction of intracellular HBV DNA by all four vectors, with the miHBV7 vector again showing the least pronounced effect (Fig 4D). Congruent with the *in vivo* experiment in Fig 3C, there was a trend towards better HBV inhibition with the TuD co-expressing vector as compared to the shHBV7-only construct, most noticeable in the HBV RNA analyses (Fig 4C).

**Long-term *in vivo* side-by-side comparison of all four strategies**

We next extended the *in vivo* analysis of the shHBV7 versus shHBV7/TuDHBV7 vectors to 3 months and included the Ago-2 co-expression and miRNA embedding strategies to directly compare their efficiency and safety. All vector variants were intravenously injected into HBV-transgenic mice at $1 \times 10^{11}$ particles per animal (Fig 5A). Congruent with our short-term *in vivo* analysis (Fig 3C, also confirmed in Fig 6A), the shRNA-only vector potently knocked down HBV, culminating in 97% serum HBsAg reduction at 4 weeks post-injection (Fig 5B, purple; corresponding HBV DNA and RNA data are shown in Fig EV2). However, this vector also induced toxicity as evidenced by a retarded gain in body weight of the treated animals (compared to all other shHBV7 expression strategies, Fig 5C) and elevated ALT levels at 1–4 weeks post-injection (Fig 5D). This coincided with a reduction of liver mass in individual

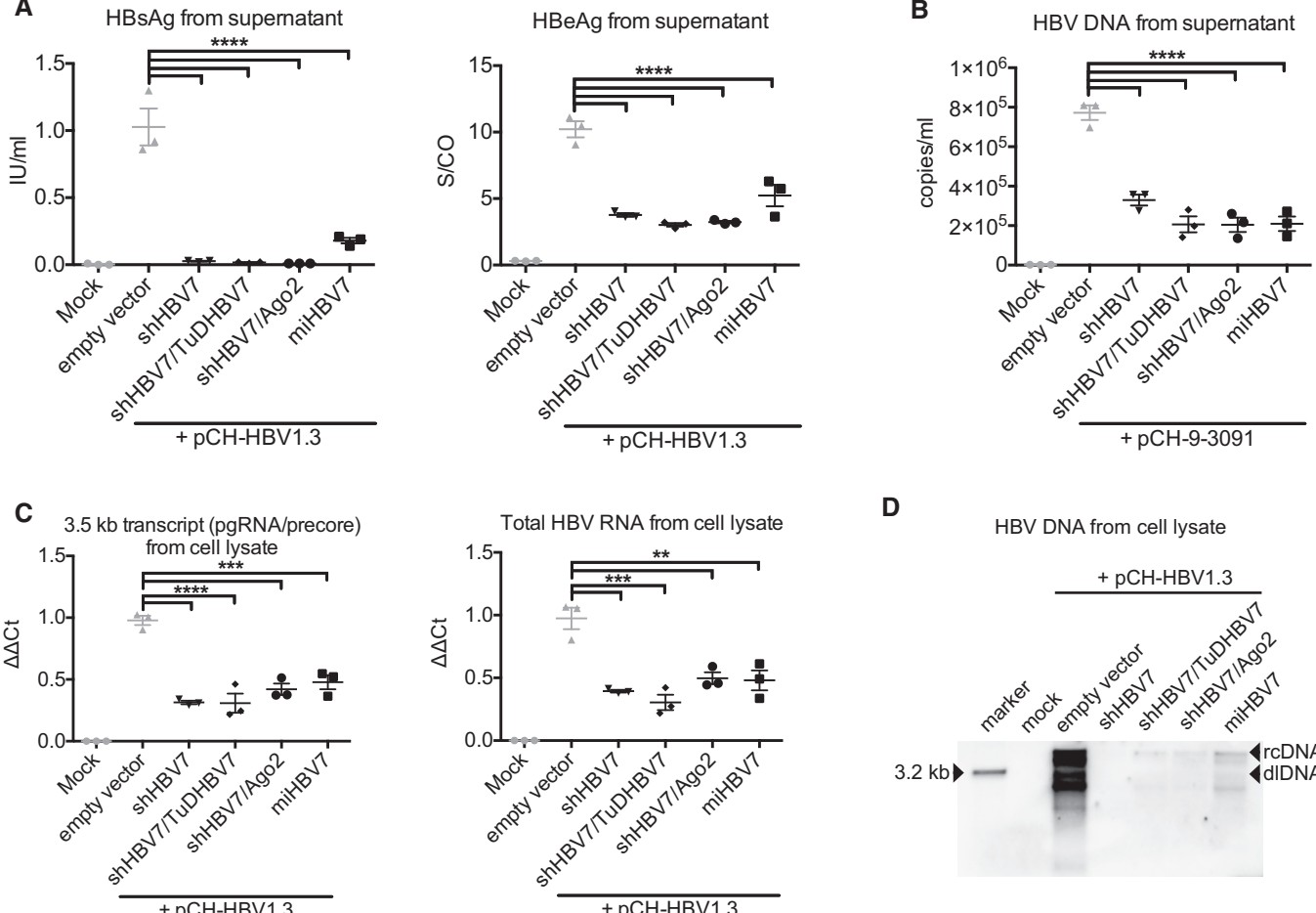

**Figure 4. *In vitro* comparison of different RNAi triggers.**

A  Huh7 cells were co-transfected with the HBV expression plasmid pCH-HBV1.3 and the four principal constructs examined in this study, or with an empty AAV expression plasmid that served as negative control. Another negative control was mock-transfected cells. HBeAg and HBsAg were quantified in the supernatant 48 h later.

B  Measurement of HBV DNA in cell culture supernatants 4 days after co-transfection of Huh7 cells with the plasmid encoding the 1.1-fold over-length HBV sequence (pCH-9-3091) and the same RNAi expression plasmids as in panel (A). PCR was performed using primers that discriminate between HBV rcDNA and the 1.1-fold over-length HBV sequence contained in the plasmid.

C  RT–qPCR results using primers specific for the 3.5-kb HBV transcript (including pre-genomic RNA and the pre-core transcript; left), or primers that detect all HBV transcripts (right).

D  Southern blot analysis of lysates obtained 72 h post-transfection of HepG2 cells with the same plasmids as described in panel (A).

Data information: Diagrams in panels (A–C) show mean values and SEM (all experiments were performed in triplicates). Significance was calculated using one-way ANOVA with Tukey's multiple comparison correction. S/CO, signal-to-control ratio; n.s., non-significant; rcDNA, relaxed circular DNA; dlDNA, duplex-linear DNA; *P < 0.05; **P < 0.01; ***P < 0.001; ****P < 0.0001. See Appendix Table S1 for exact *n*- and *P*-values.

mice treated with the shHBV7-only vector on day 15 (measured in the separate experiment shown in Fig EV3, purple). The same side effects—reduced body weight and ALT elevation—were also observed with an unrelated control shRNA (Fig 5C and D, black), reaffirming reports (Grimm *et al*, 2006; Beer *et al*, 2010) that even low levels of shRNA are not fully devoid of detectable side effects.

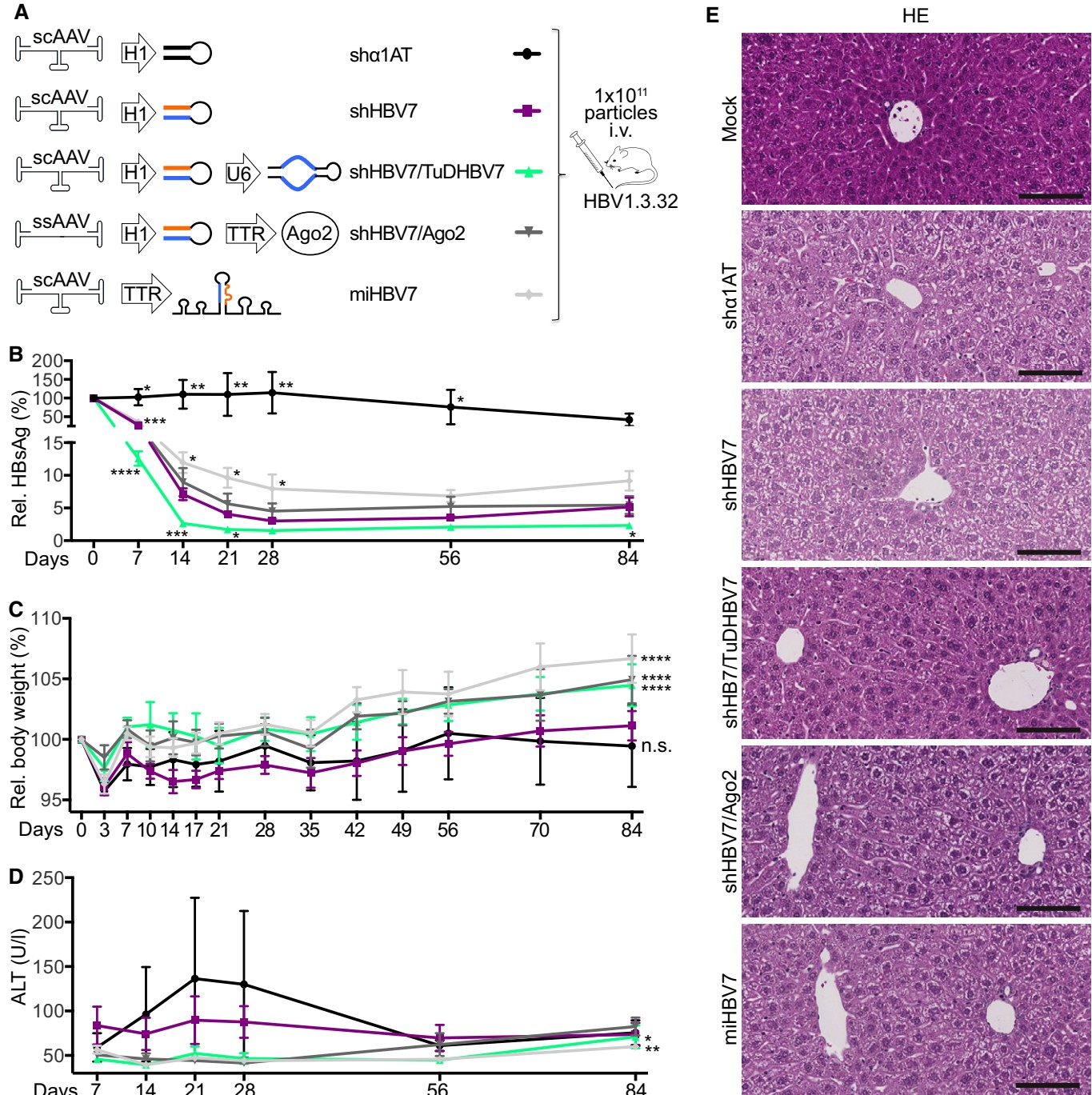

**Figure 5.   Long-term analysis of different anti-HBV AAV vectors in HBV-transgenic mice.**

A     Schemes of the five different vectors used for the *in vivo* comparison. ss, single-stranded; sc, self-complementary; i.v., intravenously.

B–D  Determination of anti-HBV efficacy and safety by measuring HBsAg in the serum of treated mice (B), their body weight (C) and serum ALT levels (D).

E     Haematoxylin/Eosin (HE) stains of livers harvested on day 84. Scale bar at bottom right represents 100 μm.

Data information: Diagrams show mean values and SEM from 5 to 6 mice per group. Significance in panel (B) was calculated using Student *t*-test, and in panels (C and D) using one-way ANOVA, comparing each group to the vector expressing only shHBV7. n.s., non-significant; *$P < 0.05$; **$P < 0.01$; ***$P < 0.001$. See Appendix Table S1 for exact *n*- and *P*-values.

Important to note is that the observed toxicity was overall mild, as shown by the absence of significant histological changes at late (day 84, Fig 5E) or early time points (day 15, Fig EV4) post-AAV injection. Together, this underscores that even presumably optimised conditions—weak shRNA promoter and moderate vector dose—cannot fully overcome low-level acute toxicity from traditional shRNA vectors and that additional improvements are needed.

Remarkably, co-expression of Ago-2, embedding of shHBV7 in a miRNA context as well as co-expression of TuDHBV7 were all able to rescue these side effects. Treated mice gained body weight (Fig 5C), showed no elevation of ALT activity (Fig 5D) and no reduction of liver mass at day 15, identical to control animals treated with a non-coding AAV (Fig EV3). This also verified that toxicity from the shRNA-only vector was exclusively due to the shRNA expression and not to any other vector component. The safety improvements over the shRNA-only vector tended to be best in mice where the shRNA was expressed from the miR-122 scaffold and under the liver-specific TTR promoter, in line with reports that miRNA embedding reduces *in vivo* RNAi toxicity (Ely *et al*, 2008, 2009; McBride *et al*, 2008; Boudreau *et al*, 2009) (Figs 5C and D, and EV3). However, the differences to the Ago-2 or TuD co-expression strategies were not statistically significant. Moreover, of all configurations, the TTR-miHBV7 vector was least efficient at inhibiting HBV (Fig 5B), consistent with our cell culture data (Fig 4).

Interestingly, efficient long-term HBV inhibition was obtained with the vector co-expressing Ago-2 from the liver-specific TTR promoter, to a degree matching the conventional shRNA-only construct (Fig 5B). Considering that the Ago-2 vector was a single-stranded AAV which expresses more slowly and less potently than scAAV genomes that were used for the shRNA-only vector, this is remarkable and validates our previous finding that Ago-2 co-delivery boosts shRNA potency in mouse livers (Grimm *et al*, 2010; Borner *et al*, 2013). In addition, over-expressing Ago-2 relieved toxicity as compared to mice receiving only the shRNA, as evidenced by weight gains and normal ALT levels in the shHBV7/Ago-2 cohort (Fig 5C and D).

Noteworthy, the strongest and most persistent HBV knock-down was achieved with the bi-cistronic vector co-expressing shHBV7 and the sense-strand TuD, which inhibited HBsAg by 98.5% and significantly outperformed all other constructs at early and late time points (Fig 5B, green). As observed for Ago-2 co-expression and miRNA strategies, the shHBV/TuDHBV7 vector was able to alleviate *in vivo* RNAi toxicity as documented by measurements of body weights (Fig 5C), serum transaminases (Fig 5D) and liver masses (Fig EV3). Furthermore, histological analysis of livers neither revealed significant changes in liver microarchitecture nor liver damage when compared to a mock control (Fig 5E). This superior antiviral efficacy combined with an excellent safety profile prompted us to select the shHBV/TuDHBV7 vector as our prime candidate.

## Analysis of global gene expression in vector-treated mice

Next, we assessed whether expression of the shRNA alone had influenced liver gene expression in treated animals, and whether the results differed after TuD co-delivery. To this end, we again injected HBV-transgenic mice with the same vectors used in Fig 5 and this time also included an empty AAV vector as control. As in our previous experiment, the shHBV7/TuDHBV7 vector was faster and more efficient at suppressing HBV than all other vectors, achieving HBsAg reductions of 96.7% 15 days after injection (compared to 93.5% in the shHBV7-only group) (Fig 6A, same animals as in Fig EV3). We harvested all livers at this time point (day 15) and profiled more than 26,000 genes of four animals per treatment group. As shown in Fig 6B, expression of shHBV7 alone had caused significant dysregulation of 126 genes (67 down- and 59 up-regulated) compared to control animals. Amongst the 67 down-regulated genes, we could analyse the sequences of 51 (Fig 6C; Appendix Tables S2 and S3). Interestingly, 41 (80.4%) of these carried a shHBV7 sense-strand seed match (nt 2–7), representing a statistically significant enrichment (compared to the background frequency; two-sided chi-square test, $P < 0.05$). Twenty-two of the 51 genes (43.1%) had an antisense-strand seed match (21 carried both). Only 9 of the 67 down-regulated genes lacked a seed match for either of the two shHBV7 strands. Amongst the 59 up-regulated genes, a sense-strand seed match was found in 66.0% and an antisense-strand seed match in 32.0%. Both numbers reflect the background frequency of all genes on the microarray chip (sense: 65.8%; antisense 31.2%), suggesting no specific correlation between shHBV7 and the up-regulated genes.

Remarkably, this perturbation of gene expression was relieved in all three cohorts treated with the improved shRNA vectors, as evidenced by only one (shHBV7/TuD) or no (shHBV7/Ago-2 or miHBV7) significantly dysregulated genes (Fig 6B). Overall, this transcriptome pattern correlated well with the toxicity data (Fig 5C and D) and liver weight measurements (Fig EV3), where the shHBV7 group was likewise the only noticeable outlier.

We then performed the reverse analysis, that is instead of identifying genes in each vector group that were significantly dysregulated, we first categorised all genes based on the presence of shHBV7 seed matches (for each shRNA strand) and subsequently compared their degree of alteration. For this, we focused on the 3′UTR sequences which we could retrieve for nearly 19,000 genes. Interestingly, we found approximately five times more shHBV7 sense- than antisense-strand seed matches (1,933 versus 388; Appendix Table S2). As expected, we observed no changes in genes lacking shHBV7 seed-strand seed matches in their 3′UTRs, when comparing the different experimental groups to the empty vector control (Fig 6D top). Also as predicted, genes harbouring antisense-strand seed matches were suppressed across all groups to a similar extent, indicating that all vector configurations can mediate antisense-strand-mediated off-targeting (Fig 6D bottom). Of note, genes carrying sense-strand seed matches were suppressed in all groups with the sole exception of animals that had co-expressed the shRNA sense-strand TuD, which significantly differed from all other groups (Fig 6D middle).

Finally, we directly compared gene expression between shHBV7- and shHBV7/TuDHBV7-treated animals. We used the cumulative distribution function (CDF, see Materials and Methods) to investigate whether genes with shHBV7 seed matches in their 3′UTR were differentially influenced by TuDHBV7 co-expression compared to genes without the respective seed match. Remarkably, CDF analysis showed that genes with a 3′UTR shHBV7 sense-strand seed match were significantly ($P < 0.05$, two-sided, two-sample Welch's *t*-test) more up-regulated by TuDHBV7 co-expression than genes without this match (Fig EV5). No such correlation was observed for genes

    

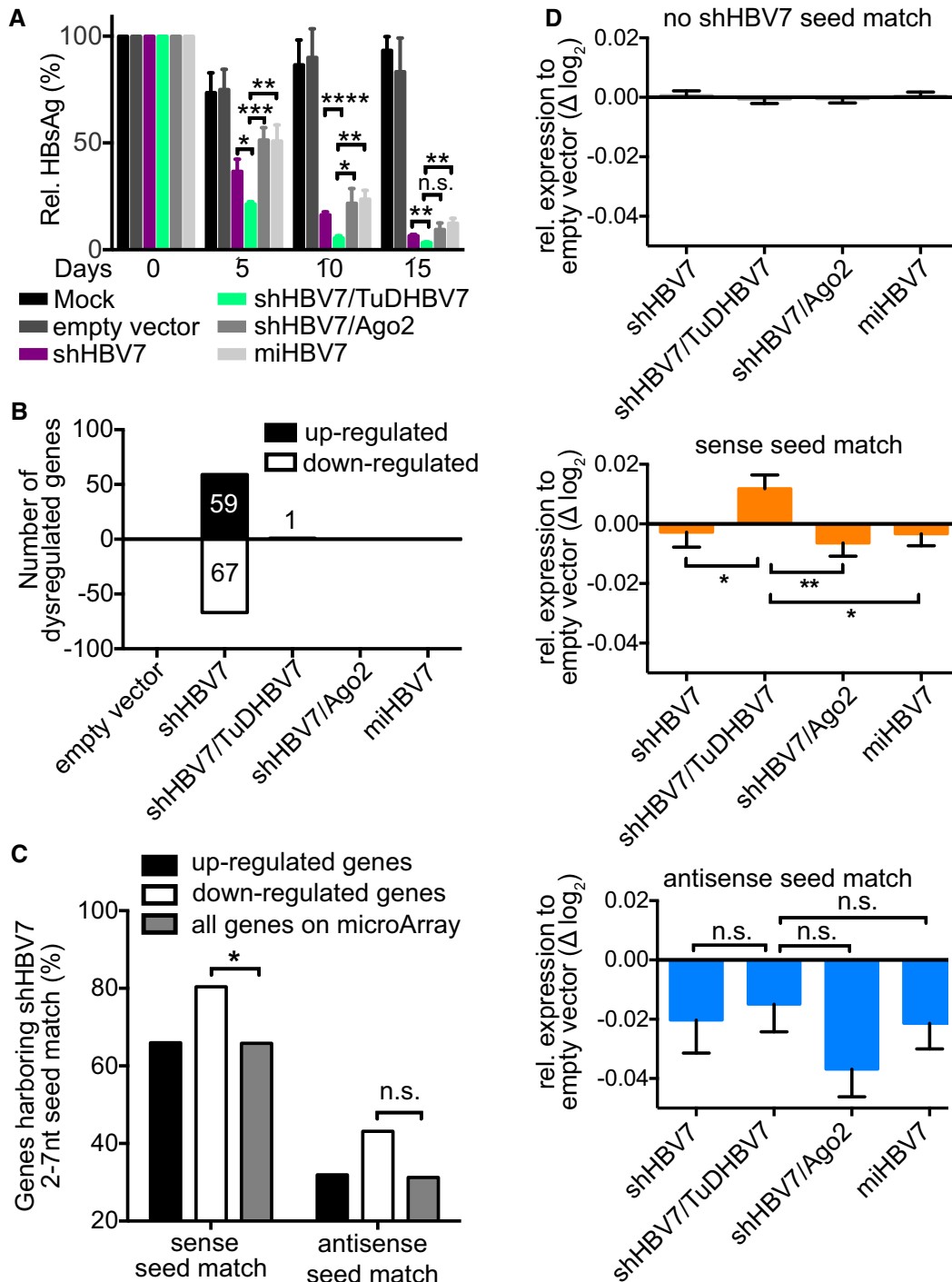

**Figure 6.  Short-term *in vivo* study to analyse shHBV7 off-target activity.**

A   HBV-transgenic mice (five in mock group, six in all others) were injected intravenously with $1 \times 10^{11}$ particles of the shown AAV8 vectors, and HBsAg was measured from serum at various time points post-injection. Fifteen days post-treatment, mRNA expression in livers was analysed using microarrays. Significance was calculated using Student *t*-test.

B   Genes significantly dysregulated compared to mock treated animals (false discovery rate < 0.25; compared to mock).

C   Transcripts of genes significantly up- or down-regulated by shHBV7 were analysed for the presence of shHBV7 sense or antisense 2- to 7-nt seed matches and compared to the background frequency within all genes on the microarray. Significance was calculated using chi-square test with Yates' correction.

D   All genes on the microarray were grouped according to the presence of 3′UTR shHBV7 sense or antisense seed matches. Shown is their relative expression compared to the empty vector. Significance was calculated using Mann–Whitney test.

Data information: Diagrams in panels (A and D) show mean values and SEM n.s., non-significant; \*P < 0.05; \*\*P < 0.01. See Appendix Table S1 for exact *n*- and *P*-values.

                    

harbouring (or not) a 3′UTR shHBV7 antisense-strand seed match (not shown), indicating that the TuD had specifically rescued genes with a sense-strand seed match, as hoped for.

## Discussion

The aim of our study was to evaluate the potential of a novel RNAi expression strategy that we have recently established in cultured cells (Mockenhaupt *et al*, 2015)—co-expression of shRNA together with an inhibitor of its sense-strand activity—for therapeutic suppression of HBV in livers of adult HBV-transgenic mice. Our key findings were (i) that the dual shRNA/TuD AAV vector gave robust and long-term *in vivo* HBV inhibition without detectable toxicity and (ii) that it overall outperformed three alternative RNAi expression strategies including conventional shRNA, together implying its great promise as a clinically applicable new anti-HBV modality (summarised in Table 1).

To develop this novel vector, we built on the prior identification of challenges for the translation of antiviral RNAi strategies. One major issue is concern about the specificity of gene inhibition and competition with the endogenous miRNA machinery (Grimm *et al*, 2006; Diederichs *et al*, 2008; Khan *et al*, 2009; Grimm, 2011). Although these effects are shRNA dose-dependent, it is imprudent to reduce vector amounts as it implies a risk of missing HBV-infected cells and/or of achieving inadequate knock-downs. Indeed, when we used a dose of $5 \times 10^{10}$ AAV particles, liver transduction and inhibition of HBV replication in HBV-transgenic mice were below 50% (not shown). A twofold increase to $1 \times 10^{11}$ boosted anti-HBV potency to over 90%, but this subtle change also induced toxicity, congruent with data with other shRNAs delivered at the same AAV8 dose in mice (Grimm *et al*, 2006; Maczuga *et al*, 2014) and illustrating the difficulties in striking the balance between potency and safety when targeting HBV *in vivo*.

A second challenge with HBV and other viruses is their existence as different genotypes and their propensity for mutational escape under therapy. This creates a pressure to target regions that not only permit potent and specific RNAi but are also highly conserved, to ensure broad and persistent efficacy. There are two possible solutions which are both inherently problematic. One is rational selection of new shRNAs that are potent and conserved; yet, this may collide with the desire to also reduce off-target or saturation liabilities. An example is shHBV7 whose target is preserved and which is effective, but not safe when expressed alone. Alternatively, users may already have a pre-validated and robust shRNA, but this may again be suboptimal with respect to safety. Both scenarios thus necessitate *de novo* design and screening of additional shRNAs, as current state-of-the-art technologies offer no means to directly optimise the safety of an existing RNAi hairpin.

This problem is also not solved by one of the three approaches that we evaluated here, shRNA embedding in an artificial miRNA context, as it likewise requires redesigning available shRNAs. Still, congruent with prior work (McBride *et al*, 2008; Boudreau *et al*, 2009), miHBV7 was one of our safest vectors. Furthermore, it is the only strategy of the three allowing RNAi expression from RNA polymerase II promoters which provides opportunities for spatio-temporal control, as exemplified here with the liver-specific TTR promoter. However, the miHBV7 vector was less efficient than the conventional shRNA or the other two expression strategies (Ago-2 or TuD), confirming *in vivo* data with similar constructs (Boudreau *et al*, 2008; Ely *et al*, 2008) and further illustrating the competition between potency and safety.

**Table 1. Overview over evaluated AAV-based RNAi expression strategies.**

| Vector | scAAV8-H1-shHBV7 | scAAV8-H1-shHBV7-U6-TuDHBV7 | ssAAV8-H1-shHBV7-TTR-Ago2 | scAAV8-TTR-miHBV7 |
|---|---|---|---|---|
| AAV genome | Self-complementary | Self-complementary | Single-stranded | Self-complementary |
| Liver-specific expression | No | No | No/Yes[a] | Yes |
| HBsAg knock-down (%) | | | | |
| Maximum[b] | 97.0 | 98.5 | 95.5 | 93.2 |
| Week 12[b] | 94.8 | 97.7 | 94.6 | 90.8 |
| ALT elevated[c] | Yes | No | No | No |
| Body weight improved[d] | N/A | Yes | Yes | Yes |
| Liver mass reduced[e] | Yes | No | No | No |
| Dysregulated genes[f] (relative to mock; adj. *P* < 0.25) | 126 | 1 | 0 | 0 |
| shRNA off-target activity | Sense and antisense | Antisense | Sense and antisense | Sense and antisense |

Shown in the columns are the four RNAi expression strategies that were studied and compared here: (i) shRNA alone, (ii) shRNA plus TuD, (iii) shRNA plus Ago-2 and (iv) shRNA in a miRNA context (from left to right). Rows summarise key features and findings from their evaluation in cells and mice in this study. N/A, not applicable.
[a]The shRNA was under the control of the ubiquitous H1 promoter, whereas Ago-2 was expressed from the liver-specific TTR promoter.
[b]Data in Fig 5B.
[c]Data in Fig 5D.
[d]A "Yes" indicates a relative improvement of body weight as compared to the shHBV7-only group (data in Fig 5C).
[e]Data in Fig EV3.
[f]Data in Fig 6B.

In this regard, a major advantage of the other two approaches—Ago-2 or TuD co-expression—is that both are fully compatible with existing or new shRNAs. For the TuD strategy, the only requirement is that the shRNA follows our simple design rules (Mockenhaupt *et al*, 2015). Both approaches thus typically permit to boost shRNA potency and safety without a need to redesign the RNAi trigger. Intriguingly, despite their different modes of action, both strategies similarly improved shRNA toxicity. In the case of Ago-2 over-expression, a likely explanation is that more RISC complexes became available for shHBV7 and endogenous miRNAs, thus alleviating a central bottleneck in the RNAi pathway (Diederichs *et al*, 2008; Grimm *et al*, 2010; Cuccato *et al*, 2011; Borner *et al*, 2013). Importantly, deliberate Ago-2 co-expression from a shRNA-encoding AAV vector as a means to improve anti-HBV RNAi *in vivo* was never reported before.

The same is true for our original strategy that we tested here, for the first time, in cellular and murine HBV models, that is counteraction of shRNA sense-strand activity with TuDs. One key asset is that its only prerequisite is location of the shRNA sense strand in the 5′ arm (Mockenhaupt *et al*, 2015), which is typical for published shRNAs and a default in most shRNA design algorithms. Further remarkable is that this strategy combines high efficiency with low toxicity, yielding the fastest, strongest and most persistent *in vivo* HBV knock-down in all our comparisons. Our present and recent data (Mockenhaupt *et al*, 2015) collectively suggest that these benefits reflect a synergism of direct and indirect effects. One indirect consequence of shRNA sense-strand inhibition could be that it frees RISC for loading of the desired antisense strand and subsequent target mRNA degradation, which can explain the better anti-HBV efficiency. Support comes from Jin *et al* who separately expressed the two strands of a siRNA and noted that a relative reduction of sense strands increased RISC loading with the intended antisense strand (Jin *et al*, 2012). The additional RISC could also be utilised by cellular miRNAs which would in turn mitigate competition with the exogenous shRNAs and hence alleviate toxicity caused by saturation of the RNAi pathway. This would further contribute to long-term *in vivo* RNAi safety and potency, akin to the effect of Ago-2 over-expression (see above) and also consistent with our mouse data.

Furthermore, a direct consequence of TuD co-expression that should additionally boost shRNA safety may be partial reversal of adverse off-targeting by the shRNA sense strand. This is implied by our cell culture data where the TuD strongly inhibited shHBV7 sense-strand activity towards a cognate reporter, consistent with our recent independent *in vitro* observations with other AAV/TuD vectors (Mockenhaupt *et al*, 2015). The new *in vivo* data further support this conclusion, as the group of genes harbouring shHBV7 sense-strand seed matches in their 3′UTR were suppressed by all RNAi constructs except for the shHBV7/TuDHBV7 vector. Our notion that genes with antisense-strand seed matches were suppressed to a fivefold higher extent can be explained by a dilution effect from high target abundance (Arvey *et al*, 2010), considering that shHBV7 sense-strand seed matches were five times more frequent in the entire gene pool in the mouse liver. Further evidence for TuD-mediated *in vivo* prevention of sense-strand off-targeting comes from our direct comparison of gene expression between shHBV7- and shHBV7/TuDHBV7-treated animals using CDF, showing that genes with 3′UTR sense-strand seed matches were

significantly more up-regulated by TuDHBV7 than genes without. Finally, reduced toxicity and off-target activity are further reflected by the 126 dysregulated genes in our shHBV7-only mouse cohort, versus one or none in all other groups including TuD-treated animals. The number of 126 presumed off-targets is highly reminiscent of results by Maczuga *et al* (2014) who described 106 dysregulated genes in mice treated with an AAV/shRNA vector, indicating this may be a typical range *in vivo*. Still, for reasons explained in detail in the Appendix Supplementary Discussion, one should exert caution when interpreting such complex *in vivo* data and assigning genes as direct off-targets.

In conclusion, we have identified a new AAV-based RNAi expression strategy that provides better *in vivo* potency, specificity and safety of HBV inhibition than traditional shRNA vectors, making it an interesting candidate for continued development towards clinical application. Particularly encouraging is the exceptional overall safety profile of AAV vectors in humans that was observed in over 100 clinical trials thus far, including numerous applications of different viral serotypes in the liver (Manno *et al*, 2006; Nathwani *et al*, 2011, 2014; D'Avola *et al*, 2016). In line with this, the first gene therapy product approved in the Western hemisphere, Glybera, is based on recombinant AAV vectors of serotype 1 (Salmon *et al*, 2014). Also important to note is that a recent report of a possible association of AAV with liver cancer (Nault *et al*, 2015) is irrelevant for the use of AAV vectors, since the specific viral sequences that were found to be integrated in the cancerous tissue are absent in AAV vectors. Moreover, the data and conclusions in this report have come under intense scrutiny within the gene therapy community (Berns *et al*, 2015; Buning & Schmidt, 2015). Further encouraging with respect to clinical translation of our strategy is that the vector doses we applied here in mice—$1 \times 10^{11}$ particles per animal, corresponding to $5 \times 10^{12}$ particles per kg—are well within the range of doses that have already been used for liver gene transfer in humans, such as $2 \times 10^{12}$, $3 \times 10^{12}$ or $1.8 \times 10^{13}$ AAV particles per kg (Nathwani *et al*, 2011, 2014; D'Avola *et al*, 2016; High & Anguela, 2016). Notably, these trials have also provided clear evidence that a single AAV vector dose can suffice to mediate stable transgene expression in the human liver for at least 4 years, in the absence of severe adverse events (High & Anguela, 2016). In addition, we can readily foresee that required vector doses in humans will further drop as an increasing number of superior—more efficient and more specific—AAV capsids are molecularly evolved in liver cells and may enter the clinic, such as AAV-DJ (Grimm *et al*, 2008) or AAV-LK03 (Lisowski *et al*, 2014). Of note, such synthetic AAV capsids can also be selected for low cross-reactivity with neutralising antibodies against natural AAV serotypes, thus permitting vector re-administration in case the effect of a single dose may wane (Grimm & Zolotukhin, 2015).

Concurrently, we are optimistic that stable over-expression of Ago-2 or of TuDs should be well tolerated in humans. Their safety is implied by the sum of data in the present work and in previous literature, including reports by us and others that Ago-2 can be persistently expressed in mammalian cells and mouse livers, without inducing abnormalities or gross pathologies (Diederichs *et al*, 2008; Grimm *et al*, 2010; Borner *et al*, 2013). Likewise, we found no evidence in the current study for adverse *in vivo* effects despite a 3-month over-expression of Ago-2, based on normal ALT levels, gains in body weight as expected as well as normal liver histology.

While we are similarly hopeful that TuD expression will be tolerated and safe in humans, we wish to point out that TuDs, in principle, may interfere with nuclear export of cellular miRNAs. This is because of their hairpin structure with a 3′ overhang, which makes them substrates for the nucleocytoplasmic transporter Exportin-5 (Bak *et al*, 2013) that is also used by miRNAs. It is further possible that long-term and high-level expression of Ago-2 may eventually dysregulate the processing and/or activity of endogenous Ago-2-dependent small RNAs. We thus consider it mandatory for follow-up work to thoroughly investigate the potential sequelae of persistent Ago-2/TuD expression, including global small RNA profiling in treated mouse livers, as further important pre-clinical steps towards the clinical evaluation of our novel concepts in HBV patients. This future work should also comprise a transfer of our vectors and strategies to alternative animal models, such as the hepadnavirus-infected woodchuck or mice with "humanised" livers, that will allow to study the impact of our vectors on HBV cccDNA persistence and ideally also on HBV-specific immunity and carcinogenesis.

Finally of note, the high versatility and easy customisation for any shRNA of interest renders our concept highly intriguing not only for improvement of HBV therapeutics. Instead, our recent cell culture data already imply its great potential to also enhance vector-based RNAi strategies against HCV (Mockenhaupt *et al*, 2015), and we can readily envision similar uses to advance treatment options for numerous other infectious or genetic diseases of the liver and further organs.

# Materials and Methods

### Plasmids

The plasmids expressing shRNAs shHBV4 to 7 that were used in the cell culture studies were cloned by direct insertion of the respective shRNA-encoding oligonucleotides (Appendix Table S5) into a self-complementary AAV vector plasmid previously reported by us (Grimm *et al*, 2006), containing an H1 promoter followed by two *BbsI* sites for oligonucleotide insertion as well as an RSV promoter-driven *gfp* reporter. For *in vivo* studies, we eliminated this reporter cassette to avoid artefacts from promoter interference and possible toxicities from the expressed GFP protein. We therefore replaced the RSV promoter in one of our existing plasmids (encoding a 19-mer shRNA against human alpha-1-antitrypsin, α1AT) (Grimm *et al*, 2006) with a SV40 polyadenylation signal that we PCR-amplified from the psiCheck2 vector (Promega, Mannheim, Germany) and cloned using *SalI*/*BamHI* sites, resulting in plasmid pBS-H1-shα1AT. Next, we exchanged the H1-shα1AT cassette with a PCR-amplified H1-shHBV7 fragment using *AscI*/*XhoI* sites, to obtain an anti-HBV shRNA vector lacking the RSV-*gfp* reporter. An empty control vector not encoding any transcript was generated by replacing the H1-shα1AT cassette with the bovine growth hormone polyadenylation (bGH/polyA) signal, likewise via *AscI*/*XhoI* digestion.

The single-stranded AAV vector plasmid pSSV9-H1-shHBV7-TTR-Ago2 was produced in two steps, by (i) exchanging the CMV promoter in our previously published vector (Grimm *et al*, 2010) with a PCR-amplified TTR promoter fragment (via *SpeI*/*SacI*) and by (ii) inserting the PCR-amplified H1-shHBV7 cassette (via *AscI*/*XhoI*).

The pri-miHBV7-encoding fragment was produced by overlap-extension PCR, using primers a/b and c/d (Appendix Table S5) in two separate PCRs and by performing a third PCR using primers a/d and a mixture of the products from the first two PCRs. The final product was then inserted behind the TTR promoter in a double-stranded AAV vector plasmid using *NheI*/*XhoI* sites.

TuDs were produced by running a PCR with the respective oligonucleotides (without further template) and insertion of the PCR product into an empty U6-TuD expression vector (Mockenhaupt *et al*, 2015) using unique *BsmBI* sites. The H1-shRNA expression cassette was exchanged with H1-shα1AT/shHBV7 using *AscI* and *XhoI*. For *in vitro* validation of TuDHBV7, oligonucleotides containing binding sites (Appendix Table S5) for the shHBV7 sense or antisense strand were inserted into the psiCheck2 plasmid using *XhoI*/*NotI* sites in the 3′UTR of the *Renilla* reporter. Plasmids pCH-HBV1.3 and pCH-9-3091 used in the HBV cell culture studies both express HBV wild-type genotype D. Plasmid pCH-HBV1.3 contains a 1.3-fold over-length HBV genome, whereas the one in pCH-9-3091 is 1.1-fold over-length.

### Cell culture experiments

All cell culture experiments except those conducted for subsequent Southern blot analyses (see below) were performed with Huh7 cells. Mycoplasma contamination was excluded for all cell lines. Huh7 cells were grown in DMEM medium supplemented with 10% foetal bovine serum, 50 U/ml penicillin/streptomycin, 2 mM L-glutamine, 1% sodium pyruvate and 1% MEM non-essential amino acids (all Life Technologies, Carlsbad, CA, USA) and kept at 37°C in humidified incubators at 5% $CO_2$. For the shRNA selection and comparison of different shRNA expression strategies, Huh7 cells were grown in 24-well plates and co-transfected (three wells per group) with pCH-HBV1.3 and the shRNA expression plasmid using Fugene HD (Promega, Mannheim, Germany). Instead of pCH-HBV1.3, plasmid pCH-9-3091 was used in experiments aimed at measuring HBV DNA in the cell culture supernatant (Figs 2B and 4B). For the Southern blot experiments in Figs 2D and 4D, HepG2 cells grown in 6-well cell culture dishes were transfected with the same protocol. One day after transfection, the cells were washed with PBS, supplied with fresh media containing 2.5% DMSO and harvested after additional 72 h. For the shRNA screen, 183 ng of both HBV- and shRNA-expressing plasmids was co-transfected per well. For a fair comparison of the different shHBV7 expression strategies, the transfected DNA amount was adjusted to the molecular weight of the respective plasmid in order to deliver equal numbers. After 48 h, supernatants were collected and centrifuged at 5,000 *g* for 5 min, before HBV antigen levels were determined using the HBsAg quantitative test and the HBeAg test on Architect (Abbott Laboratories, Abbott Park, Illinois, USA). To analyse HBV DNA in supernatant, a DNA digest was performed to remove plasmid DNA by adding 20 U/ml DNase I (Roche Diagnostics, Mannheim, Germany). After incubation at 37°C for 2 h, the reaction was stopped by adding EDTA to a final concentration of 8 mM before nucleic acid was extracted as described below.

For *in vitro* validation of TuDHBV7, activities of the shHBV7 sense and antisense strands were measured using the dual-luciferase reporter psiCheck2 (Promega) engineered to contain binding sites for one of the two shRNA strands. Therefore, Huh7 cells grown in

96-well plates were co-transfected with 10 ng of psiCheck2 and 100 ng of the plasmids expressing shRNA and TuD. Cells were harvested 48 h after transfection into lysis buffer supplied in the Dual-Glo luciferase kit according to the manufacturer's protocol (Promega), and *Renilla* and Firefly luciferase activities were quantified using a GloMax96 microplate luminometer (Promega). Relative knock-downs were determined by using the group co-transfected with the luciferase reporter without binding site and the shα1AT/TuDα1AT plasmid as reference.

### AAV vector production

AAV vectors were produced using a standard protocol involving triple transfection of 293T cells with equal amounts of an AAV8 helper plasmid (encoding AAV2 *rep* and AAV8 *cap* genes), an AAV vector plasmid (encoding the transgene[s]) and an adenoviral helper construct (Grimm, 2002). The AAV vector plasmids were either based on an optimised self-complementary AAV vector genome that we reported before (Grimm *et al*, 2006) or, for co-expression of shHBV7 and Ago-2, the conventional pSSV9 construct whose packaging results in single-stranded DNA-containing AAV particles (Samulski *et al*, 1987). Vectors were produced as described previously (Borner *et al*, 2013) and purified using a caesium chloride ultracentrifugation gradient. Briefly, after removal of high molecular weight impurities (addition of 1/39 volume of 1 M $CaCl_2$, incubation for 1 h on ice, centrifugation at 10,000 *g* for 15 min and transfer of supernatant) and PEG precipitation of AAVs (addition of 1/4 volume of 40% PEG8000/2.5 M NaCl, overnight incubation on ice and centrifugation at 2,500 *g* and 4°C for 30 min), the AAV-containing pellet was resuspended in 10 ml Na-Hepes resuspension buffer (50 mM HEPES, 0.15 M NaCl, 25 mM EDTA). After adding 13.2 g caesium chloride and adjusting the optical density (refractive index, RI) to 1.3710 using a refractometer, ultracentrifugation was performed for 22 h at 208,000 *g* (Type Ti 70 rotor, Optima L 90 K ultracentrifuge; both Beckman Coulter, Krefeld, Germany) and 21°C. Fractions with RIs of 1.3711–1.3766 were collected and dialysed against phosphate-buffered saline (overnight at 4°C, four times exchange of PBS) using a Slide-A-Lyzer G2 dialysis cassette with a 20K molecular weight cut-off (Thermo Fisher Scientific Inc., Rockford, IL, USA). AAV vectors were quantified by TaqMan PCR on a Rotor-Gene 6000 (Qiagen, Hilden, Germany) using the Sensimix II Probe kit Mastermix (Bioline, London, UK). All reactions were performed in triplicates in a 10 μl final volume. To ensure equal quantification of AAV vectors using eGFP and TTR primers (Appendix Table S5), the same plasmid standard as well as the same AAV positive control (each containing the eGFP and TTR sequences) was used in both assays.

### Mouse experiments

HBV-transgenic mice used in this study (HBV1.3.32) express replication-competent wild-type HBV genotype D (Guidotti *et al*, 1995). Animals were maintained according to the guidelines of local authorities (Government of Upper Bavaria, Germany), and all experiments were approved by them. AAV vectors or 0.9% saline for the mock controls were injected intravenously into the tail vein, and blood was collected via retroorbital or facial vein bleeding. Upon termination of the experiment, mice were euthanised using carbon

dioxide, and blood and organs were dissected for further analysis. Blood was centrifuged at 5,000 *g* for 10 min, and serum ALT activity was measured in a 1:4 dilution in PBS using the Reflotron GPT/ALT test (Roche Diagnostics). HBsAg was measured in a 1:30 dilution in PBS using the HBsAg quantitative test on Architect and HBeAg in a 1:20 dilution in PBS using the HBe 2.0 test on Axsym (both from Abbott Laboratories). For pathohistological analysis, sections (2 μm) of livers (fixed in 4% paraformaldehyde and paraffin-embedded) were stained with haematoxylin/eosin. For analysis, slides were scanned using a SCN 400 slide scanner (Leica).

### Animal inclusion criteria and cohort assignments

Only male mice from 2 to 4 months of age were used in this study. Statistical advice was obtained in which the study was defined as exploratory orientation study, and six animals per group were deemed sufficient in order to gain reliable estimates of quantitative effect sizes. Animals were bled 1 day before treatment and allocated into groups with equal HBsAg, HBeAg, age and body weight. Treatment groups were not blinded.

### Nucleic acid extraction, PCR and Southern blot analysis

RNA from cultured cells was extracted with the MN Nucleo Spin RNA kit (Macherey Nagel, Duren, Germany), and cDNA synthesised with the Superscript III kit (Thermo Fisher Scientific). HBV transcripts were amplified with primers specific for only the 3.5-kb transcripts, or with primers binding to the common 3′ end of all HBV transcripts (Yan *et al*, 2012). Beta-2-microglobulin (B2M) was used as reference gene for cell culture experiments. To measure HBV DNA in cell culture supernatants, DNA was extracted from 400 μl supernatant using the Abbott mSample Preparation System DNA on the M24SP machine (both Abbott). The PCR was performed with primers "HBV rcDNA selective" which preferentially amplify the HBV rcDNA rather than the 1.1-fold over-length HBV genome in the pCH-9-3091 plasmid. All PCRs were performed on a LightCycler 480 (Roche Diagnostics) using the primers and PCR conditions shown in Appendix Table S5. For the Southern blot analyses, cells were lysed and treated with DNase I and RNase A for 3 h at 37°C. Cytoplasmic capsids were precipitated with polyethylene glycol (PEG8000) and digested using sodium dodecyl sulphate and proteinase K. After 3-h incubation at 37°C, capsid-associated DNA was purified by phenol–chloroform extraction, followed by ethanol precipitation. Viral DNA was separated on a 1.3% agarose gel, transferred to a nylon membrane and UV cross-linked. The membrane was then hybridised with digoxigenin-labelled HBV-specific probe at 65°C overnight, and HBV DNA was visualised using the DIG Luminescent Detection kit according to the manufacturer's instructions (Roche).

### Transcriptome analysis

Isolated liver pieces were conserved in RNAlater (Qiagen) and then lysed using TissueLyser LT (Qiagen), before whole RNA was extracted using the miRNeasy kit including on-column DNA digest with the RNase-Free DNase set (both Qiagen). After assessing the RNA quality using a Bioanalyzer 2100 (Agilent, Santa Clara, CA, USA), cDNA was generated from four mice per treatment group

using the Ambion WT Expression kit (Life Technologies, Carlsbad, CA, USA). The cDNAs were fragmented and labelled with the Affymetrix GeneChip WT Terminal Labeling Kit before running them on GeneChip Mouse Gene 2.0 St Arrays (Affymetrix, Santa Clara, CA, USA).

## Statistical analyses

Microarray data were analysed in the "R" software environment (http://cran.r-project.org) using the Limma (Smyth, 2005) and the oligo package (Carvalho & Irizarry, 2010) from Bioconductor (http://bioconductor.org). Raw expression measurements have been quantile-normalised and background-corrected using the RMA background correction. A fixed-effects linear model was fit for each individual gene to estimate pairwise expression differences. Using a principal component analysis and pairwise Pearson correlation coefficients for quality control of overall performance of the arrays, four samples were found to show different expression levels than the remaining arrays. Thus, they were excluded from further analysis as outlined in the metadata table of the Gene Expression Omnibus database file (accession number is given below). Empirical Bayes approach was used to moderate the standard errors of the normalised $\log_2$ fold changes. Two-sided moderated paired t-statistics and log-odds of differential expression (B statistics) as well as raw and adjusted *P*-values (false discovery rate) controlled by Benjamini–Hochberg were computed to identify differentially expressed genes. The absolute $\log_2$ fold change > 1 and a corrected *P*-value smaller than the testing level (alpha) of 0.25 were defined as significantly differentially expressed genes. The microarray data are available in the Gene Expression Omnibus database (GEO accession number GSE72335; http://www.ncbi.nlm.nih.gov/geo/query/acc.cgi?acc = GSE72335).

Data of mouse and cell culture experiments were tested with D'Agostino-Pearson omnibus test for Gaussian distribution. If data were not normally distributed, Mann–Whitney test was used to test for significant differences. *F*-test was used to test whether variances of groups were similar. If variances were found to differ significantly, *t*-test with Welch's correction was used.

## Seed match analysis

3′UTR and whole transcript sequences of all genes represented on the microarray for which an Ensembl ID was available were downloaded from www.ensembl.org/biomart and screened for 2- to 7-nt seed matches of the shHBV7 sense and antisense strand. In cases where several variants per gene existed, the transcript with the highest number of seed matches was chosen. Genes with several seed matches were counted only once, and genes which were represented on the array more than once were excluded from the analysis to prevent bias. To compare relative gene expression of genes depending on 3′UTR shHBV7 seed matches (Fig 6D), gene expression was normalised to all genes for which a 3′UTR seed match analysis was available. Statistical analysis was performed with unpaired *t*-test using Welch's correction if applicable. We computed the cumulative distribution function (CDF) to directly compare gene expression depending on the presence of shHBV7 seed matches between shHBV7 and shHBV/TuDHBV7 treated animals. For this, the relative expression of genes with a 3′UTR seed match was tested

**The paper explained**

**Problem**

Hepatitis B virus (HBV) is a notorious human pathogen that is present in roughly 250 million humans and is responsible for 686,000 annual deaths. The success of this virus is due to the limitations of standard-of-care antiviral therapy, including the need for daily and lifelong application, high costs, failure to eliminate the virus or to inhibit expression of HBV antigens, as well as risks of side effects and of emergence of resistant HBV mutants. A possible solution is RNA interference (RNAi), a powerful tool for targeted mRNA silencing that can be directed against cellular or foreign sequences involved in human diseases including viral transcripts. To this end, however, clinical translation of RNAi technologies urgently requires further improvements in *in vivo* efficiency, specificity and safety.

**Results**

In this paper, we systematically evaluated three different strategies for expression of anti-HBV short hairpin RNA (shRNA, an RNAi trigger), using recombinant Adeno-associated virus 8 (AAV8) vectors for delivery to livers of adult immunocompetent HBV-transgenic mice. Briefly, we (i) embedded the shRNA in a microRNA scaffold under a liver-specific promoter; (ii) co-expressed Argonaute-2, a rate-limiting RNAi factor; or (iii) co-delivered a decoy ("TuD") against the shRNA sense strand, to alleviate off-target gene regulation. Notably, while a conventional shRNA vector caused *in vivo* toxicity including weight loss, liver damage and dysregulation of > 100 genes in the liver, the three other strategies minimised all these adverse side effects. Best results were obtained with the new AAV8 vector co-expressing anti-HBV shRNA and TuD which gave the most robust and most persistent HBV knockdown (> 98% HBsAg suppression for at least 12 weeks).

**Impact**

We have identified a new AAV-based RNAi expression strategy that provides better *in vivo* potency, specificity and safety of HBV inhibition than traditional shRNA vectors or than two alternative approaches. Importantly, because our concept is highly versatile and easy to customise, it is not restricted to HBV but instead also promising and interesting for a large variety of other human diseases that are vulnerable to RNAi. For instance, we have recently demonstrated its potential to likewise enhance vector-based RNAi strategies against hepatitis C virus, and we can readily envision its future development towards many other clinical applications for infectious or genetic diseases of the liver and further human tissues.

for significant difference against the relative expression of genes without a seed match. A two-sample, two-sided Welch's *t*-test was used and an alpha level of 0.05.

## Pathway analysis

Pathway enrichment of the genes identified to be significant was obtained by querying the DAVID database (Jiao *et al*, 2012) for GO BP/5, GO BP/FAT and KEGG, Reactome, BBID, BioCarta and Panther pathway annotation using the R-package RDAVIDWebService (Fresno & Fernandez, 2013). If identical annotation terms were returned for GO BP/5 and GO BP/FAT, only the one with the lowest EASE *P*-value was retained (Appendix Table S4).

## Target conservation

To analyse conservation of targeted HBV regions, reverse complements of shRNA antisense strands were aligned with representative

sequences of all HBV genotypes derived from the HBV regulatory sequence database HBV RegDB (http://lancelot.otago.ac.nz/HBV RegDB) (Panjaworayan *et al*, 2007) using Mega version 6.06 for Mac (Tamura *et al*, 2013). The phylogenetic tree of HBV sequences in Appendix Fig S1C was generated using the neighbour-joining method in Mega.

**Expanded View** for this article is available online.

## Acknowledgements

The authors thank Frank Chisari for providing the HBV1.3.32 mouse line. We moreover appreciate technical support from Ellen Wiedtke, Silvia Weidner and Ruth Hillermann, and we are grateful for help with graphic design from Laura Michler. Furthermore, we thank Anne-Kathrin Herrmann and further members of our laboratories for critical reading of the manuscript. SG, SM and DG acknowledge support by the Cluster of Excellence CellNetworks (German Research Foundation [DFG], EXC81) and the Collaborative Research Center SFB1129 (DFG, DG). UP and DG were supported by the Transregional Collaborative Research Center TRR179 (DFG, TP18) and the Helmholtz Initiative on Synthetic Biology. MH is supported by an ERC CoG Grant (HepatoMetaboPath). TM received a Clinical Leave stipend by the Academy of the German Center for Infection Research (DZIF). BK is supported by the German Research Foundation (SPP1395/InKoMBio Busch 900/6-1).

## Author contributions

TM, SM, UP and DG designed research. TM, SG, SM and DG designed and created AAV vectors. TM and SG performed cell culture experiments. TM and NR performed mouse experiments. CK conducted Southern blot analyses. FS and TM analysed seed match frequency. BK analysed microarray data. TM and MH analysed histological stainings. TM, UP and DG wrote the manuscript. All authors critically read and revised the final manuscript for important intellectual content prior to submission.

## Conflict of interest

The authors declare that they have no conflict of interest.

## For more information

Microarray data are available in the Gene Expression Omnibus database; accession # GSE72335: www.ncbi.nlm.nih.gov/geo/query/acc.cgi?acc=GSE72335

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
