## [Review Process File · EMBO Molecular Medicine]

Blocking sense-strand activity improves potency, safety and specificity of anti-hepatitis B virus short hairpin RNA

Thomas Michler, Stefanie Große, Stefan Mockenhaupt, Natalie Röder, Ferdinand Stückler, Bettina Knapp, Chunkyu Ko, Mathias Heikenwalder, Ulrike Protzer, Dirk Grimm

Corresponding author: Dirk Grimm, University of Heidelberg

Review timeline:

Submission date:	27 December 2015
Editorial Decision:	10 February 2016
Revision received:	05 June 2016
Editorial Decision:	20 June 2016
Revision received:	03 July 2016
Accepted:	05 July 2016

Transaction Report:

Editor: Céline Carret

1st Editorial Decision

10 February 2016

Thank you for the submission of your manuscript to EMBO Molecular Medicine. I am really sorry that it has take so long to get back to you, in part due to festive season. We have now heard back from the two referees who we asked to evaluate your manuscript. Although the referees find the study to be of potential interest, they also raise a number of concerns that must be addressed in the next final version of your article.

You will see from the comments below, that both referees found the study novel and interesting. However, they both suggest providing additional explanations and details. They also recommend detailing how the data could be translated into clinical application, which is an important concept for us. Referee 1, in addition, suggests a couple of additional experiments that would strengthen the data (points 8 and 9).

Given these evaluations, I would like to give you the opportunity to revise your manuscript, with the understanding that the referees' concerns must be fully addressed and that acceptance of the manuscript would entail a second round of review. Please note that it is EMBO Molecular Medicine policy to allow only a single round of revision and that, as acceptance or rejection of the manuscript will depend on another round of review, your responses should be as complete as possible.

Please read below for important editorial formatting.

I look forward to receiving your revised manuscript.

***** Reviewer's comments *****

Referee #1 (Remarks):

General comments

1. The technical quality of the manuscript is high. The cooperation of groups with excellent qualification in the fields of AAV, HBV and genome analysis has led to a convincing combination of different experimental systems. However, the analysis of HBV expression is not completely satisfactory. Some quantitations of HBV markers are suboptimal (specific point 7) or missing at all (points 8, 9).
2. Quality of evidence. The experimental evidence for the conclusions drawn is strong (but see specific points 8, 9).
3. Novelty. The approach is novel and probably a big step forward in decreasing the toxicity of the shRNA therapy while increasing its efficacy.
4. Medical impact. Both the use of AAV vectors for gene delivery as well as RNAi for suppressing unwanted gene activity have been in the focus of modern molecular medicine. But both approaches are hampered by technical limitations and severe side effects. The idea to suppress statistically generated side effects of the sense component in shRNAs is innovative and according to the results very promising. HBV is a highly relevant target and the design of the shRNA and the TuD component is convincing and may lead to new therapeutic approach in humans. But the limitations and remaining or new dangers have to be addressed in the discussion in more detail (specific points 10, 11).
5. Adequacy of the model system. The work includes HBV expressing cell cultures and more importantly transgenic mice which express HBV and its by-products HBsAg and HBeAg. The animal model is very relevant because it can indicate the toxicity of the therapeutic agent and the residual replication capacity of HBV. The shHBV7/TuD construct is very convincingly shown to exert optimal HBV antigen suppression at undetectable hepatic pathogenicity. But the effect on HBV replication is not described which would have been obvious (specific point 9). A transfer to a more authentic animal system like the hepadnavirus-infected woodchuck should be discussed. The perspectives how the approach could be applied in the clinic should be discussed (point 10).
6. Clarity and interest for non-specialist. The text is very clear and interesting concerning the design of shRNAs and supporting elements like TuD. But the role of the approach for the control of the HBV infection is not optimally described (specific points 1-5) and the term TuD is not explained at all (Specific point 6).

Specific points

1. The title is somewhat enigmatic to the non-specialized reader. Blocking instead of neutralization as in the short title may be more appropriate.
2. Introduction. More recent and possibly more accurate references on the worldwide number of chronic HBsAg carriers and the estimated annual death toll caused by HBV are: Lancet. 2015;Vol. 386, No. 10003, p.1546-55 and Lancet 2015;Vol. 385, No. 9963, p117-171.

3. Introduction. Current interventions with NAs fail entirely to inhibit the viral transcription and translation including core/pol and X products. Interferon may induce impaired expression of HBV proteins but is very ineffective.

4. Introduction. HBeAg and HBsAg are indeed key players in the immune modulation caused by HBV, but HBx may have an important effect on the immune response as well and contribute to development of HCC. Core and Pol maintain the first step of replication and mediate formation of resistance or escape mutants.

5. Introduction. The authors may briefly explain what they consider a "functional cure" and possibly give a reference for that term (see also point 10b).

6. Introduction. Tough decoys (TuD) were originally designed for knockdown of miRNAs. The term may not be known to everybody. The authors may briefly mention who coined this term: Haraguchi et al. NAR 2009; 37(6):e43.

7. Methods p. 24. Neither the Enzygnost HBeAg test nor the AxSym HBsAg test is designed to provide quantitative results. Extinctions (Enzygnost) or S/CO values are only semi-quantitative parameters and follow a sigmoid curve which leads to under-estimates at high concentrations and over-estimates at low concentrations. I believe that the results would be even better if a true quantitation would have been done. The authors should mention in the result section that the percentages of reduction do not refer to true concentrations but only to ELISA signals.

8. Results. The therapeutical approach targets the HBV RNAs, but the authors measure only levels of secreted HBsAg and HBeAg.

a. Quantitative assay of HBV mRNAs and pregenome RNA is tedious and needs cell extracts or liver tissue, but at least in one experiment it should have been checked whether the levels of secreted antigen do indeed reflect the activity of the shRNA on the corresponding mRNAs.

b. Alternatively, it could be tested whether the cell media and/or the sera from the HBV mice contain HBV RNA packed in exosomes, core or immature HBV particles and whether these RNA levels change with therapy.

9. Results. The authors focused on "knockdown" of HBsAg and HBeAg. But the HBV production and secretion would be the most important marker. Their shRNA/TuD approach should be at least as active on virus formation because their shRNA targets the pregenomic RNA as well which is at same time the mRNA for pol and core. Thus, the effect on HBV production would be threefold. On the other side may the pregenomic RNA be rapidly packaged by the pol/core complex in cis and become inaccessible for the shRNA. Therefore, the activity on the pregenomic RNA and HBV core particle formation cannot be deduced from the HBeAg secretion. But the authors do not provide data on this.

a. They should have done a quantitative PCR for HBV in the supernatant of HBV expressing cells and in the sera of the transgenic mice and may consider to study this parameter in at least some experiments.

b. One could argue that NA therapy would be better for suppression of replication but inhibiting pregenomic RNA, polymerase and core would prevent formation of new progeny with potential resistance. Furthermore, long-term NA therapy may have side effects.

10. Discussion. The authors do not discuss the perspectives how to transfer their approach to human patients.

a. One point is the dosage of AAV-vectors. They need a minimum dose of 10E11 particles per mouse since every hepatocyte has to be reached by the vector. Extrapolated to humans one would need at least 10E14 particle per dosage. Is this feasible at reasonable cost and in view of potential toxicity? How often would the treatment have to be repeated?

b. How can the approach lead to the functional cure and eliminate or inactivate the cccDNA?

11. Discussion. Recently appeared an article: Nault et al. "Recurrent AAV2-related insertional mutagenesis in human hepatocellular carcinomas." *Nature Genetics* 47, 1187-1193 (2015). The authors describe insertions of AAV2 DNA next to oncogenic genes in HCC patients. Is this a problem for the use of AAV vectors?

12. Overall, the figures are well designed.

- a. But they are too small in the pdf and I wonder how they will look online on a small tablet.
- b. Too many panels are packed together in one figure.
- c. Dots with standard deviations would be more informative than solid bars with asterisks referring to significance.

13. Supplemental figure 2 parts A and B are inconsistent and partly wrong. Part A shows 5 transcription start sites whereas part B shows only 4 co-linear mRNAs.

- a. Part B shows two mRNAs covering the entire genome: pre-core and pre-genome RNA but part A shows only one arrowhead. There should be two closely spaced start sites.
- b. Part A shows correctly three mRNAs for the three co-carboxyterminal HBsAg proteins (L-, M-, and S-HBs), but part B shows only one mRNA.
- c. The core mRNA encodes and expresses also the HBV polymerase. Thus, I suggest to add pol after core

Referee #2 (Comments on Novelty/Model System):

The manuscript described three different strategies to express anti-HBV small RNAs in a preclinical HBV mouse model. Using rAAV as gene transfer vector, the authors embedded the shRNA into miR-122 scaffold, co-expressed Ago-2 with shRNA, or co-delivered a TuD with shRNA to improve the potency, safety and specificity of anti-HCV shRNA. The authors demonstrated that all three strategies minimized the adverse side effects as compared to a conventional shRNA vectors and the co-delivery of a TuD is the best for efficacy and persistence of HBV knockdown, showing substantial promise for clinical translation. Overall, the manuscript is well written and the data are clearly presented. If the authors can address some points listed below, this reviewer recommends the editor to consider a revised version of the paper for publication in *EMBO Mol. Med.*

Referee #2 (Remarks):

The manuscript described three different strategies to express anti-HBV small RNAs in a preclinical HBV mouse model. Using rAAV as gene transfer vector, the authors embedded the shRNA into miR-122 scaffold, co-expressed Ago-2 with shRNA, or co-delivered a TuD with shRNA to improve the potency, safety and specificity of anti-HCV shRNA. The authors demonstrated that all three strategies minimized the adverse side effects as compared to a conventional shRNA vectors and the co-delivery of a TuD is the best for efficacy and persistence of HBV knockdown, showing substantial promise for clinical translation. Overall, the manuscript is well written and the data are clearly presented.

Specific points:

1. In Fig 2C, no statistical significance was found between shHBV7 and shHBV/Ago2.
2. In the four tested shRNAs against HBV, the most potent one, shHBV7, showed strong sense strand activity (Fig 3B). It may be more informative for the authors to compare the sense strand activity of shHBV7 with its gene silencing potency (Fig 2B) side-by-side.
3. TuDHBV7 was used to abolish the sense strand activity of shHBV7, but could the TuD should also target the sense binding sites in the luciferase mRNA 3'UTR as well? The data in Fig 3B showed no reduction of luciferase when TuDHBV7 was co-transfected with sense bs. Could the authors explain this selective inhibition of TuDHBV7?

4. shRNA plus the TuD is supposed to minimize the sense strand activity, but could authors explain why such a combination achieved the best RNAi potency in mice (Fig 3C)?
5. In the long-term study (Fig 4), co-expression of Ago-2 or TuD RNA with shRNA improved the safety of AAV-mediated shRNA delivery in mice. But is it possible that long-term Ago-2 over-expression may cause global elevations of miRNA activity, alterations in overall gene expression and unpredicted phenotype outcomes? TuD RNA construct was designed to transport the TuD RNA transcripts efficiently from nuclear to cytoplasm where the mature miRNAs located. Saturation of Exportin-5 is one cause of shRNA toxicity. Is the transportation of TuD RNA to cytoplasm independent of Exportin-5? The shRNA plus TuD strategy is supposed to be improved on specificity, Could authors explain how it can improve the safety of shRNA? In other words, the authors should discuss the connection between the sense strand neutralization and the improvement of shRNA potency and safety.
6. It seems to be a surprise to see almost no dysregulated genes in shHBV/TuDHBV, shHBV7/Ago2 and miHBV treated mice (Fig 5B). Co-expression of Ago-2 is supposed to increase the small RNA activity including both sense and antisense strand. Some unprocessed sense strand can also be possibly derived from the miHBV expression cassette. Did the authors sequence the liver small RNAs to show the abundance of the sense strand RNA?
7. The authors should clarify in the discussion section how to translate co-delivery of Ago-2 or TuD with shRNA into clinical applications.

1st Revision - authors' response

05 June 2016

*Referee #1 (Remarks):**General comments*

1. *The technical quality of the manuscript is high. The cooperation of groups with excellent qualification in the fields of AAV, HBV and genome analysis has led to a convincing combination of different experimental systems. However, the analysis of HBV expression is not completely satisfactory. Some quantitations of HBV markers are suboptimal (specific point 7) or missing at all (points 8, 9).*
2. *Quality of evidence. The experimental evidence for the conclusions drawn is strong (but see specific points 8, 9).*
3. *Novelty. The approach is novel and probably a big step forward in decreasing the toxicity of the shRNA therapy while increasing its efficacy.*
4. *Medical impact. Both the use of AAV vectors for gene delivery as well as RNAi for suppressing unwanted gene activity have been in the focus of modern molecular medicine. But both approaches are hampered by technical limitations and severe side effects. The idea to suppress statistically generated side effects of the sense component in shRNAs is innovative and according to the results very promising. HBV is a highly relevant target and the design of the shRNA and the TuD component is convincing and may lead to new therapeutic approach in humans. But the limitations and remaining or new dangers have to be addressed in the discussion in more detail (specific points 10, 11).*
5. *Adequacy of the model system. The work includes HBV expressing cell cultures and more importantly transgenic mice which express HBV and its by-products HBsAg and HBeAg. The animal model is very relevant because it can indicate the toxicity of the therapeutic agent and the residual replication capacity of HBV. The shHBV7/TuD construct is very convincingly shown to exert optimal HBV antigen suppression at undetectable hepatic pathogenicity. But the effect on HBV replication is not described which would have been obvious (specific point 9). A transfer to a more authentic animal system like the hepadnavirus-infected woodchuck should be discussed. The perspectives how the approach could be applied in the clinic should be discussed (point 10).*
6. *Clarity and interest for non-specialist. The text is very clear and interesting concerning the design of shRNAs and supporting elements like TuD. But the role of the approach for the control of the*

HBV infection is not optimally described (specific points 1-5) and the term TuD is not explained at all (Specific point 6).

We are very grateful to this supportive and positive reviewer for repeatedly highlighting the quality, novelty and innovative nature of our work. We are equally thankful for the critical comments that we have fully addressed during the revision of our work and that have helped us to significantly improve our manuscript. As requested, this includes an extended Discussion covering the limitations and remaining/new dangers of our approaches, as well as an outlook how our vectors could be applied in the clinic. Please see our responses to the specific points below for more details on the individual improvements that we have made.

Specific points

1. The title is somewhat enigmatic to the non-specialized reader. Blocking instead of neutralization as in the short title may be more appropriate.

We thank the reviewer for this helpful suggestion and have changed the title to "Blocking sense strand activity improves potency, safety and specificity of anti-hepatitis B virus short hairpin RNA".

2. Introduction. More recent and possibly more accurate references on the worldwide number of chronic HBsAg carriers and the estimated annual death toll caused by HBV are: Lancet. 2015;Vol. 386, No. 10003, p.1546-55 and Lancet 2015;Vol. 385, No. 9963, p117-171.

We are grateful to the reviewer for pointing this out and have gladly added the suggested reference Schweitzer *et al.*, Lancet 2015, Vol. 386, p. 1546-55 as well as MacLachlan *et al.*, Lancet 2015, Vol. 386, p. 1515-17. Furthermore, we have updated the numbers of HBV cases in the Introduction based on these newly added references.

3. Introduction. Current interventions with NAs fail entirely to inhibit the viral transcription and translation including core/pol and X products. Interferon may induce impaired expression of HBV proteins but is very ineffective.

The reviewer is of course correct. In our original Introduction, we had specifically focused on HBsAg and HBeAg. Based on the reviewer's helpful comment, we have now expanded this part and also mention HBV core/pol and X. Please also see next comment and response.

4. Introduction. HBeAg and HBsAg are indeed key players in the immune modulation caused by HBV, but HBx may have an important effect on the immune response as well and contribute to development of HCC. Core and Pol maintain the first step of replication and mediate formation of resistance or escape mutants.

We thank the reviewer for again highlighting important aspects of HBV pathobiology that we have gladly considered while refining our Introduction.

5. Introduction. The authors may briefly explain what they consider a "functional cure" and possibly give a reference for that term (see also point 10b).

The use of the term "functional cure" was indeed imprecise, and we thank this reviewer very much for pointing this out to us. We aimed to state that suppression of all viral transcripts and proteins - in contrast to NA therapy - should prevent all adverse effects of HBV infection, including immunomodulation, as well as direct carcinogenic and chronic inflammation. However, "functional cure" is often defined as anti-HBs seroconversion which we do not expect after RNAi therapy. To prevent any misunderstanding, we have thus removed the term from the revised manuscript.

*6. Introduction. Tough decoys (TuD) were originally designed for knockdown of miRNAs. The term may not be known to everybody. The authors may briefly mention who coined this term: Haraguchi *et al.* NAR 2009; 37(6):e43.*

We thank the reviewer for highlighting this shortcoming. While we had referenced our own recent publication where we had employed TuDs (Mockenhaupt *et al.*, PNAS 2015, 112, E4007-E4016), it is of course important to give credit to the original inventors of this technology. Accordingly, we have now added a statement that TuDs were initially developed by Haraguchi and colleagues to silence cellular miRNAs, before we found that they can be repurposed to also selectively bind and inactivate shRNA sense strands.

7. Methods p. 24. Neither the Enzygnost HBeAg test nor the AxSym HBsAg test is designed to provide quantitative results. Extinctions (Enzygnost) or S/CO values are only semi-quantitative parameters and follow a sigmoid curve which leads to under-estimates at high concentrations and over-estimates at low concentrations. I believe that the results would be even better if a true quantitation would have been done. The authors should mention in the result section that the percentages of reduction do not refer to true concentrations but only to ELISA signals.

The reviewer correctly points out that the Enzygnost HBeAg and AxSym HBsAg tests are not designed to provide quantitative data. Therefore, we have now used the Architect quantitative HBsAg chemiluminescent micro particle immunoassay also for our cell culture experiments. Indeed, as predicted by the reviewer, we measured even stronger reductions than previously observed and shown in the original manuscript, where we had used the AxSym HBsAg test. We have thus gladly included the new quantitative HBsAg data in new Figures 2 and 4, which replace the former non-quantitative data in Figure 2C. We thank the reviewer for this helpful comment and for allowing us to show even better data in our revised manuscript.

To our best knowledge, a quantitative HBeAg test is not commercially available. For the revised manuscript, we have now used the Architect HBeAg test instead of the Enzygnost HBeAg test for the cell culture experiments (since Architect allows to determine both, HBsAg and HBeAg, from the same sample). We are aware that also this test under-estimates HBeAg at high concentrations and over-estimates at low concentrations. Importantly, however, we have measured serial dilutions to identify the linear range of this assay. The range in which the cell culture supernatants were tested positive (between 1 and 12 S/CO) lies well within the linear range of the Architect HBeAg test, providing us with confidence in our data.

8. Results. The therapeutical approach targets the HBV RNAs, but the authors measure only levels of secreted HBsAg and HBeAg.

a. Quantitative assay of HBV mRNAs and pregenome RNA is tedious and needs cell extracts or liver tissue, but at least in one experiment it should have been checked whether the levels of secreted antigen do indeed reflect the activity of the shRNA on the corresponding mRNAs.

b. Alternatively, it could be tested whether the cell media and/or the sera from the HBV mice contain HBV RNA packed in exosomes, core or immature HBV particles and whether these RNA levels change with therapy.

We fully agree with the reviewer that it is also interesting to study HBV RNA in addition to measuring HBV antigen levels; only the latter was shown in the original manuscript. As requested by the reviewer, we have therefore performed the suggested experiments. Specifically, we now show qRT-PCR-based quantifications of the 3.5 kb HBV transcripts as well as of total HBV RNA from cells transfected with the different anti-HBV shRNAs (new Figure 2) or the various RNAi constructs (new Figure 4). In addition, in new Figure EV2B, we also show quantifications of the 3.5 kb HBV transcripts and total HBV RNA in liver lysates from mice treated with our different AAV vectors.

Importantly, all these new analyses fully support our previous conclusions, that shHBV7 is the most robust shRNA amongst the four candidates that we have screened (new Figure 2C), and that the novel shHBV7/TuD vector tends to give the best HBV inhibition (protein and RNA; new Figure 4C). Moreover, they additionally confirm that the miRNA-based vector is least efficient amongst the four RNAi vectors, as already seen on the protein level in the original manuscript and as now independently validated by the new RNA (and also DNA, see next point) analyses in Figure 4C and Figure EV2.

9. Results. The authors focused on "knockdown" of HBsAg and HBeAg. But the HBV production and secretion would be the most important marker. Their shRNA/TuD approach should be at least as active on virus formation because their shRNA targets the pregenomic RNA as well which is at same time the mRNA for pol and core. Thus, the effect on HBV production would be threefold. On the other side may the pregenomic RNA be rapidly packaged by the pol/core complex in cis and become inaccessible for the shRNA. Therefore, the activity on the pregenomic RNA and HBV core particle formation cannot be deduced from the HBeAg secretion. But the authors do not provide data on this.

a. They should have done a quantitative PCR for HBV in the supernatant of HBV expressing cells and in the sera of the transgenic mice and may consider to study this parameter in at least some experiments.

As above for the RNA analyses, we likewise concur with the reviewer that quantification of HBV DNA would also be informative and important. Therefore, we have gladly performed the suggested experiments as well and have added the new data to the revised manuscript as new Figure 2B/D, new Figure 4B/D, and new Figure EV2A (where they complement the new RNA data, see previous point).

As in the case of the newly performed RNA analyses (see this reviewer's point #8 above), we were very pleased to find that the requested DNA analyses further confirmed the superior efficacy of the shHBV7 shRNA, as now seen in the qRT-PCR data in new Figure 2B and the Southern blot in new Figure 2D. Likewise, the new data on HBV DNA in cell supernatants (new Figure 4B) or cell lysates (new Figure 4D) confirm very robust HBV inhibition with all four vectors in cell culture. Last but not least, the new HBV DNA measurements in the serum of AAV vector-treated animals (new Figure EV2A) confirm the trends towards improved *in vivo* HBV knockdown with the TuD vector, and the lower potency of the miRNA-based vector, congruent with the RNA and protein data.

Altogether, we are thus very thankful to this reviewer for requesting the additional HBV RNA (see above) and DNA analyses, as we can now present a much more complete and consistent picture on the activity of our different RNAi vector designs *in vitro* and *in vivo*.

b. One could argue that NA therapy would be better for suppression of replication but inhibiting pregenomic RNA, polymerase and core would prevent formation of new progeny with potential resistance. Furthermore, long-term NA therapy may have side effects.

We are very grateful to the reviewer for these important thoughts. The concept that suppression of new progeny would prevent generation of resistant mutants is very appealing and suggests a further benefit of RNAi therapies over reverse transcriptase inhibitors. We also agree that long-term NA therapy may trigger side effects, such as kidney insufficiency caused by Tenofovir-Disoproxilfumarat and Entecavir. We have thus gladly stressed this point from the reviewer in the revised Introduction of our manuscript.

10. Discussion. The authors do not discuss the perspectives how to transfer their approach to human patients.

a. One point is the dosage of AAV-vectors. They need a minimum dose of $10E11$ particles per mouse since every hepatocyte has to be reached by the vector. Extrapolated to humans one would need at least $10E14$ particle per dosage. Is this feasible at reasonable cost and in view of potential toxicity? How often would the treatment have to be repeated?

The reviewer asks important questions that we gladly address below and in the Discussion of the revised manuscript.

A dose of 1×10^{11} particles in a mouse (typically 20 g) corresponds to roughly 5×10^{12} particles per kg. This is only roughly 2-fold higher than what has been used in multiple previous clinical trials with AAV8 vectors in hemophilia B patients ($2-3 \times 10^{12}$ particles per kg). Notably, it is 3.6-fold lower than the dose of 1.8×10^{13} AAV5 vector particles per kg that was applied in a most recent clinical study in patients with acute intermittent porphyria (D'Avola *et al.*, J Hepatol 2016, epub 05/24/2016, doi: 10.1016/j.jhep.2016.05.012). Accordingly, a dose of 5×10^{12} particles per kg, as it would be predicted by our study, is well within the range of what has already been tested clinically and thus feasible.

Further noteworthy is that we and many others are continuously engineering novel AAV capsid variants that outperform existing wild-type viruses including AAV8 in the liver and other organs. One recent example is capsid LK03 (Lisowski *et al.*, Nature 2014, 506, p. 382-6) that has been evolved in mice with a "humanized" liver and that outperforms AAV8 in human hepatocytes. These developments raise substantial hope that future vector/capsid generations will allow for a significant reduction of doses that need to be applied to human patients, which will in turn further enhance the feasibility of AAV/RNAi therapies.

Regarding the number of treatments - ideally, we expect that a single vector delivery will suffice to permanently block HBV expression and replication. This is based on very encouraging data in hemophilic dogs who continued to express AAV vector-encoded blood coagulation factor IX for over 10 years following AAV2 vector administration. Similarly, human hemophilic patients have expressed therapeutically relevant amounts of factor IX for up to 4 years following a single AAV8 vector dose (see new references in revised manuscript). Still, it remains possible that HBV-infected patients may ultimately require a second dose of the vector described in our paper, in case they experience liver damage which leads to cell division and thus loss of the episomal AAV vector DNA. Again, we are highly optimistic that also this hurdle will be overcome, owing to the numerous ongoing efforts to evolve new AAV capsids that are not only more efficient in the liver, but also less reactive with neutralizing anti-AAV antibodies. Consequently, such next-generation capsids can be used for a repeat AAV/RNAi administration, if needed.

We have incorporated all of these considerations into the Discussion of the revised manuscript, and we thank the reviewer for raising these points and for giving us the opportunity to improve our work.

b. How can the approach lead to the functional cure and eliminate or inactivate the cccDNA?

As stated above in our response to specific point 5, we agree with the reviewer that the term "functional cure" is misleading in our context and thus removed it from the manuscript. Elimination of cccDNA would only be possible if immunity against HBV can be restored when soluble antigens are suppressed. This, however, cannot be properly investigated without better animal models.

11. Discussion. Recently appeared an article: Nault et al. "Recurrent AAV2-related insertional mutagenesis in human hepatocellular carcinomas." Nature Genetics 47, 1187-1193 (2015). The authors describe insertions of AAV2 DNA next to oncogenic genes in HCC patients. Is this a problem for the use of AAV vectors?

The reviewer is raising another important point that was triggered by the quoted publication by Nault and colleagues, and that was subsequently widely debated in the AAV and gene therapy fields. Notably, the work by Nault *et al.* has been heavily and broadly criticized for at least four reasons. Firstly, the authors exclusively focused on wild-type AAV serotype 2, yet the wording of the original paper implies that the findings may bear relevance for recombinant AAV vectors as well. Secondly, and contributing to this confusion, the authors failed to clearly underscore that the AAV DNA sequence they found integrated in the HCC samples is not even present in recombinant AAV vectors. Thirdly, close examination of the raw data revealed that 9 out of the 11 AAV-positive HCC are actually attributable to AAV-unrelated etiologies or mutations, arguing against a role of AAV as cancer driver. Fourthly, Nault and colleagues did not detect AAV integration in the so-called AAVS1 site in chromosome 19, despite a large bulk of literature describing AAVS1 as the predominant hot spot for wild-type AAV insertion in the human genome.

For all these reasons, the paper by Nault *et al.* is not of any concern for the use of recombinant AAV vectors such as those described in our manuscript. Nonetheless, as it is likely that readers may wonder, too, we have added references to the original paper as well as to subsequent commentaries and reviews that clearly rebut the conclusions by Nault and co-workers.

12. Overall, the figures are well designed.

a. But they are too small in the pdf and I wonder how they will look online on a small tablet.

We thank the reviewer very much for the positive overall comment and for highlighting the size issue. However, we believe the small size may have been an artefact in the PDF that was generated

for the reviewers by the online submission system, since we have prepared the originals for display in 1- or 2-column width. Nonetheless, we have further improved readability wherever possible, *e.g.*, by increasing label sizes in Figures 3 and 6 (previously Figure 5). In line with the EMBO Press Figure guidelines, the font sizes used are at least size 7 (in Figure 3) or larger (all other figures). The minimum line weight used is at least 2 points. In case our manuscript will be accepted for publication, we will pay attention to further maximize size and readability of all figures within the limits permitted by *EMBO Molecular Medicine*.

b. Too many panels are packed together in one figure.

In response to this concern and also due to the addition of substantial amounts of new data, we have rearranged several panels and split up one figure. In particular, we have moved former Figure 2A to Figure 1 (new panel C), and then devoted entire new Figures 2 and 4 to former Figure 2B and 2C, respectively. We furthermore show the additional data from mouse experiments (HBV DNA from serum and HBV transcripts from liver lysates) in a new Figure EV2, rather than adding them to Figure 5 (former Figure 4), also to avoid over-loading of figures with too many panels. In case the reviewer(s) and/or editor feel(s) that additional rearrangements might be beneficial, we will of course gladly implement these when preparing the final figures (in case our manuscript will be accepted for publication).

c. Dots with standard deviations would be more informative than solid bars with asterisks referring to significance.

As requested by the reviewer, we now depict all the new data on HBV proteins, DNA and RNA shown in new Figures 2 and 4 as dots and not as solid bars. In addition, we assume that the reviewer may have also referred to the bar graphs in Figure 3C. In this case, we would prefer to keep the representation as bars instead of dots since we depict 24 groups per graph. Consequently, a representation of all these data as single dots would be too unclear and too confusing.

13. Supplemental figure 2 parts A and B are inconsistent and partly wrong. Part A shows 5 transcription start sites whereas part B shows only 4 co-linear mRNAs.

We apologize for the incorrect representation of the HBV genome, and we thank the reviewer very much for giving us the opportunity to fix our mistakes. As requested, we have made panels A and B consistent, by adding a second start site for the mRNAs covering the entire genome in panel A (see also point a below), and by adding two mRNAs in panel B (see also point b below). As a result of this correction, both panels now consistently show six HBV transcripts. Please note that due to the upgrade of several former Supplementary Figures to new Expanded View Figures, former Supplementary Figure S2 has now become Appendix Figure S1.

a. Part B shows two mRNAs covering the entire genome: pre-core and pre-genome RNA but part A shows only one arrowhead. There should be two closely spaced start sites.

The reviewer is of course correct. As requested, we have added a second start site (depicted as arrowhead, consistent with the others) to panel A.

b. Part A shows correctly three mRNAs for the three co-carboxyterminal HBsAg proteins (L-, M-, and S-HBs), but part B shows only one mRNA.

Again, the reviewer has correctly spotted a mistake. We have fixed it by now showing all three HBs transcripts - L(arge), M(iddle) and (S)mall - in the revised Appendix Figure S1B (see above, former Supplementary Figure S2B).

c. The core mRNA encodes and expresses also the HBV polymerase . Thus, I suggest to add pol after core

We thank the reviewer for this useful suggestion and have expanded the labeling in panel B as suggested.

Referee #2 (Comments on Novelty/Model System):

The manuscript described three different strategies to express anti-HBV small RNAs in a preclinical HBV mouse model. Using rAAV as gene transfer vector, the authors embedded the shRNA into miR-122 scaffold, co-expressed Ago-2 with shRNA, or co-delivered a TuD with shRNA to improve the potency, safety and specificity of anti-HCV shRNA. The authors demonstrated that all three strategies minimized the adverse side effects as compared to a conventional shRNA vectors and the co-delivery of a TuD is the best for efficacy and persistence of HBV knockdown, showing substantial promise for clinical translation. Overall, the manuscript is well written and the data are clearly presented. If the authors can address some points listed below, this reviewer recommends the editor to consider a revised version of the paper for publication in EMBO Mol. Med.

Referee #2 (Remarks):

The manuscript described three different strategies to express anti-HBV small RNAs in a preclinical HBV mouse model. Using rAAV as gene transfer vector, the authors embedded the shRNA into miR-122 scaffold, co-expressed Ago-2 with shRNA, or co-delivered a TuD with shRNA to improve the potency, safety and specificity of anti-HCV shRNA. The authors demonstrated that all three strategies minimized the adverse side effects as compared to a conventional shRNA vectors and the co-delivery of a TuD is the best for efficacy and persistence of HBV knockdown, showing substantial promise for clinical translation. Overall, the manuscript is well written and the data are clearly presented.

We are equally grateful to this second and also very positive reviewer who perfectly summarized our work and highlighted its great promise for clinical translation. We have greatly appreciated all the helpful and constructive comments by this reviewer as well, and have incorporated all of them into the revised manuscript, as detailed below.

Specific points:

1. In Fig 2C, no statistical significance was found between shHBV7 and shHBV/Ago2.

The reviewer is correct. In fact, we were not surprised by this finding. This is because we had purposely pre-selected the most potent shRNA (shHBV7) amongst the four candidates that we have tested. In our previous work (Börner *et al.*, NAR 2013, 41, e99), we had found that Ago-2 overexpression particularly boosts the knockdown efficiency of inherently weak shRNAs. Accordingly, we did not expect to find a dramatic increase of the very robust shHBV7 shRNA upon Ago-2 co-delivery. We have now re-phrased this section of the Results to state this more clearly (p. 11-12): "Despite the high inherent activity of the shHBV7 shRNA, we observed a trend towards further improvement through Ago-2 co-expression at least for the secreted parameters (best notable for HBV DNA in the supernatant, whose knockdown was increased from 57.3 to 73.5%)."

Importantly, we predicted to find a benefit from Ago-2 overexpression for the safety of anti-HBV *in vivo* RNAi. This is indeed what we observed, as shown in Figures 5 and 6 (former Figures 4 and 5). Thus, despite the absence of a statistically significant improvement of the efficacy of our potent anti-HBV shRNA, we can readily conclude that our strategy to co-express shRNA and Ago-2 is highly advantageous with respect to better clinical safety.

2. In the four tested shRNAs against HBV, the most potent one, shHBV7, showed strong sense strand activity (Fig 3B). It may be more informative for the authors to compare the sense strand activity of shHBV7 with its gene silencing potency (Fig 2B) side-by-side.

The reviewer makes an important suggestion, namely, a direct side-by-side comparison of shHBV7 sense *versus* antisense strand activity. As a matter of fact, Figure 3B already shows this comparison, since the orange bars represent sense strand activity, while the blue bars show antisense strand activity. This comparison is fairer than juxtaposing the sense strand activity in Figure 3B with the silencing potential in former Figure 2B, since the targets and assays in these two figures are different (HBV in Figure 2B, as opposed to luciferase reporter in Figure 3B). In contrast, except for the precise binding site, target and readout are identical for the comparison of shHBV7 sense *versus* antisense activity that we show in Figure 3B.

To more clearly highlight that Figure 3B already shows the comparison that this reviewer has requested, we have rephrased the corresponding sentence in the main text on p. 10 to "To assess shHBV7 sense versus antisense strand activity and TuD functionality, we co-transfected cells ...

3. TuDHBV7 was used to abolish the sense strand activity of shHBV7, but could the TuD should also target the sense binding sites in the luciferase mRNA 3'UTR as well? The data in Fig 3B showed no reduction of luciferase when TuDHBV7 was co-transfected with sense bs. Could the authors explain this selective inhibition of TuDHBV7?

The finding described by the reviewer - that the TuD itself showed no effect on the luciferase reporters (with either sense or antisense binding sites) - is exactly what we predicted and hoped for. The explanation is that the TuD hairpin functions by binding and sequestering a single-stranded RNAi molecule (in our case, the sense strand of shHBV7), but it neither enters the RNAi pathway nor does it exert knockdown activity on its own. This is a consequence of the specific TuD hairpin structure which precludes its recognition as an RNAi trigger by the cell.

In response to the reviewer's comment, we have now clarified this point by explicitly stating the lack of RNAi activity for TuDs in the sentence (p. 11): "All controls behaved as expected, since expression of the shHBV7-sensitive luciferase reporter was neither inhibited by the irrelevant shRNA nor by the two TuDs (which exert no inherent RNAi activity)."

4. shRNA plus the TuD is supposed to minimize the sense strand activity, but could authors explain why such a combination achieved the best RNAi potency in mice (Fig 3C)?

We believe this is due to a synergistic combination of direct and indirect effects, comprising the release of RISC complexes for loading of the desired antisense strand and/or cellular miRNAs (which mitigates toxicity and hence further contributes to improved long-term efficacy), as well as reversal of adverse off-targeting. We feel that we have already discussed these possibilities rather extensively on p. 20-21 as well as in the Appendix Supplementary Discussion.

Nonetheless, in response to the reviewer, we have further clarified this point by expanding the following sentences on p. 20-21 (underlined parts have been added): "The additional RISC could also be utilized by cellular miRNAs which would in turn mitigate competition with the exogenous shRNAs and hence alleviate toxicity caused by saturation of the RNAi pathway. This would further contribute to long-term in vivo RNAi safety and potency, akin to the effect of Ago-2 over-expression (see above) and also consistent with our mouse data."

5. In the long-term study (Fig 4), co-expression of Ago-2 or TuD RNA with shRNA improved the safety of AAV-mediated shRNA delivery in mice. But is it possible that long-term Ago-2 over-expression may cause global elevations of miRNA activity, alterations in overall gene expression and unpredicted phenotype outcomes? TuD RNA construct was designed to transport the TuD RNA transcripts efficiently from nuclear to cytoplasm where the mature miRNAs located. Saturation of Exportin-5 is one cause of shRNA toxicity. Is the transportation of TuD RNA to cytoplasm independent of Exportin-5? The shRNA plus TuD strategy is supposed to be improved on specificity. Could authors explain how it can improve the safety of shRNA? In other words, the authors should discuss the connection between the sense strand neutralization and the improvement of shRNA potency and safety.

The reviewer again makes a series of intriguing comments and asks important questions related to the mechanisms and benefits of our Ago-2 and TuD strategies.

Regarding Ago-2: We are confident that Ago-2 *in vivo* overexpression can be safe, based on a large amount of consistent previous data from our (D.G.) group. This comprises our 2010 publication in Journal of Clinical Investigation (Grimm *et al.*, 2010, 120, p. 3106-19) where we first demonstrated the feasibility to overexpress Ago-2 in mouse liver, in the absence of gross pathologies. It likewise includes our comprehensive 2013 study where we documented the possibility to generate stable mammalian cell lines that overexpress Ago-2, and that are largely indistinguishable from the parental cells in terms of morphology, growth behaviour, or cDNA or miRNA expression profiles (Börner *et al.*, NAR 2013, 41, e992013). Moreover, in our present work,

we also found no evidence for adverse *in vivo* effects, based on normal ALT levels, expected gains in body weight, and healthy liver histologies (Figure 5, *i.e.*, former Figure 4).

Still, in the revised manuscript, we now discuss the potential outcomes of long-term of Ago-2 overexpression in more detail. This also addresses specific point #7 by the same reviewer (see below) as well as point #10 of reviewer #1 who wondered about the perspectives how to transfer our approach to human patients.

Regarding TuDs: TuDs are indeed substrates for Exportin-5, due to their composition of a 18-bp stem region with a short 3' overhang; the latter is recognized by Exportin-5 (Bak *et al.* Mol Ther 2013, 21, p. 1478-85). Consequently, we agree with the reviewer that there is a theoretical concern about adverse saturation of this particular step in cellular RNA export. Notably, the sum of our data in the present manuscript provide no evidence that persistent TuD expression is indeed toxic, as shown in Figure 5 (former Figure 4). Nonetheless, because we appreciate the reviewer's concern, we have also included these additional considerations and references into our revised and expanded Discussion (p. 23-24). Again, this also addresses point #7 by the same reviewer (see below) and point #10 by reviewer #1 about clinical translation.

Regarding the connection between sense strand neutralization and improvement of shRNA potency and safety, we believe this is now clear in the revised Discussion. Please also see our response to point #4 by the same reviewer that highlights further amendments that we have made. On top, we have also expanded this sentence on p. 21, to even further clarify the connection between TuD activity and higher safety (underlined parts are new): "Furthermore, a direct consequence of TuD co-expression that should additionally boost shRNA safety may be partial reversal of adverse off-targeting by the shRNA sense strand."

6. It seems to be a surprise to see almost no dysregulated genes in shHBV/TuDHBV, shHBV7/Ago2 and miHBV treated mice (Fig 5B). Co-expression of Ago-2 is supposed to increase the small RNA activity including both sense and antisense strand. Some unprocessed sense strand can also be possibly derived from the miHBV expression cassette. Did the authors sequence the liver small RNAs to show the abundance of the sense strand RNA?

While we were also positively surprised by this finding, we feel very confident with these data for the reasons outlined in the paper. This includes the notable congruence with the independent recent report by Maczuga *et al.* of 106 genes (*versus* 126 here) that were dysregulated in AAV/shRNA-treated mouse livers. Even so, as described in our Appendix Supplementary Discussion, we recommend to be careful when bioinformatically analysing and interpreting such complex *in vivo* data, and in particular when assigning dysregulated genes as direct shRNA-related off-targets.

Along these lines, we fully agree with the reviewer that a comprehensive sequencing and profiling of all small RNAs in the livers of treated mice would help to better dissect such relationships. However, we also wish to point out possible limitations of this approach, including the fact that it will only reveal alterations in the steady-state level of small RNAs but not in their functionality. For instance, it would therefore not allow to detect potential off-target activity of TuDs since they bind and stably sequester small RNAs (such as the shHBV7 sense strand here), but do not necessarily affect their level in the cell. Likewise, we expect Ago-2 over-expression to predominantly enhance the activity of small RNAs, but not their expression or processing (with a very few exceptions in certain cells types, such as miR-451 which may be a direct Ago-2 substrate). These considerations are in fact well in line with our observations here (mouse liver) as well as in our 2013 study (cultured cells; Börner *et al.*, NAR 2013, 41, e992013), that persistent Ago-2 overexpression is well tolerated and leads only to marginal changes in cDNA and miRNA profiles.

In addition, we feel that a comprehensive and time-consuming small RNA sequencing including subsequent bioinformatical analysis is beyond the scope of this manuscript and the revision period. Here, our focus was to provide the first demonstration of the usefulness and clinical promise of our novel vector-mediated RNAi expression strategies for HBV therapy. We completely agree with the two reviewers that we needed to better discuss how to translate our approaches into clinical applications, and we were thus happy to revise and expand the manuscript accordingly. However, we also kindly ask for understanding that we could not concurrently perform the small RNA sequencing analysis for the present manuscript.

7. *The authors should clarify in the discussion section how to translate co-delivery of Ago-2 or TuD with shRNA into clinical applications.*

The reviewer makes a very valid point. We apologize for this shortcoming in our original manuscript and have now substantially expanded and clarified our Discussion, as requested. Please also see our responses to points #4-5 above, as well as to point #10 of reviewer #1.

2nd Editorial Decision

20 June 2016

Thank you for the submission of your revised manuscript to EMBO Molecular Medicine. We have now received the enclosed reports from the referees that were asked to re-assess it. As you will see the reviewers are now globally supportive and I am pleased to inform you that we will be able to accept your manuscript pending the following final amendments:

1) Please address the minor text change commented by referee 2. Please provide a letter INCLUDING the reviewer's reports and your detailed responses to their comments (as Word file).

Please submit your revised manuscript within two weeks. I look forward to seeing a revised form of your manuscript as soon as possible.

***** Reviewer's comments *****

Referee #1 (Comments on Novelty/Model System):

I have read the rebuttal from the authors which was very convincing and satisfied all my open questions and suggestions. The technical quality of the HBV-related data is now much improved and completely adequate. The potential translation of the therapeutic concept to the clinical level is now discussed in more detail. The text is very clear and well organized. The figures are instructive and support the conclusions.

Referee #1 (Remarks):

I thank the authors for the detailed rebuttal, the thorough revision and the addition of relevant data. The discussion covers all points of concern.

Referee #2 (Comments on Novelty/Model System):

The authors did a good job in responding to reviewer's comments and revising the manuscript

Referee #2 (Remarks):

Please clarify two minor issues below.

1. On p.49, the authors claimed " shHBV7 and pri-miR-122/HBV7 encode the identical antisense strand (blue) against HBV". where is the data in the manuscript to support the claim?
2. miHBV7 was one of the safest vectors in the study. Is it because of less sense strand activity or less RNAi potency overall from this miRNA scaffold design?

Referee #1 (Comments on Novelty/Model System):

I have read the rebuttal from the authors which was very convincing and satisfied all my open questions and suggestions. The technical quality of the HBV-related data is now much improved and completely adequate. The potential translation of the therapeutic concept to the clinical level is now discussed in more detail. The text is very clear and well organized. The figures are instructive and support the conclusions.

Referee #1 (Remarks):

I thank the authors for the detailed rebuttal, the thorough revision and the addition of relevant data. The discussion covers all points of concern.

We are very happy that we could convince and satisfy this expert reviewer, and we once again wish to express our great appreciation to her/him for the very helpful and insightful comments that helped us to substantially improve our manuscript.

Referee #2 (Comments on Novelty/Model System):

The authors did a good job in responding to reviewer's comments and revising the manuscript

We are equally grateful to this second expert who likewise supported us throughout the review process with very useful and constructive criticisms, and who thus also helped us very much to improve numerous aspects of our manuscript.

Referee #2 (Remarks):

Please clarify two minor issues below.

1. On p.49, the authors claimed " shHBV7 and pri-miR-122/HBV7 encode the identical antisense strand (blue) against HBV". where is the data in the manuscript to support the claim?

The reviewer correctly implies that we do not formally prove the expression of identical antisense strands from the two constructs. To do so would have required comprehensive small RNA sequencing analyses which, for reasons outlined in our previous responses, was beyond the scope of the manuscript and the revision period. Nonetheless, to clarify that we have intended to express identical strands, we have now amended the legends to Fig. EV1C on p. 51 to "Note that shHBV7 and pri-miR-122/HBV7 are designed to encode the identical antisense strand (blue) against HBV, ..." (changes are underlined).

2. miHBV7 was one of the safest vectors in the study. Is it because of less sense strand activity or less RNAi potency overall from this miRNA scaffold design?

We actually believe that the two possibilities raised by the reviewer are closely related, *i.e.*, the miRNA scaffold is overall less potent which then automatically translates into reduced sense strand activity. Consequently, we consider it difficult to experimentally dissect these two possibilities. Adding to the complexity is the fact that, as shown in Figure EV1C, the sense strand in the miRNA scaffold contains a few mismatches as compared to the sense strand in the shHBV7 construct. These changes were necessary to maintain and mimic the structure of the miR-122 scaffold. Importantly, the seed sequence of the two sense strands is 100% identical, as also already highlighted in Figure EV1C with the larger letters in the miHBV7 structure. Still, because of these few mismatches, we would consider a direct and quantitative experimental side-by-side comparison (using luciferase reporters as in Fig. 3) of the sense strand activities of the miRHBV7 construct with our other vectors challenging. Collectively, we believe that it is impossible at this point to conclusively distinguish the two possibilities suggested by the reviewer, and we would thus prefer to refrain from a deeper discussion of this issue in our manuscript (especially since the miHBV7 construct was not our lead candidate).

Corresponding Author Name: Dr. Dirk Grimm and Prof. Dr. Ulrike Protzer

Manuscript Number: EMM-2015-06172